# Calcium-activated 14-3-3 proteins as a molecular switch in salt stress tolerance

Zhijia Yang[1], Chongwu Wang [1], Yuan Xue[1], Xiao Liu[1], She Chen[2], ChunPeng Song [3], Yongqing Yang[1] & Yan Guo [1]

Calcium is a universal secondary messenger that triggers many cellular responses. However, it is unclear how a calcium signal is coordinately decoded by different calcium sensors, which in turn regulate downstream targets to fulfill a specific physiological function. Here we show that SOS2-LIKE PROTEIN KINASE5 (PKS5) can negatively regulate the Salt-Overly-Sensitive signaling pathway in Arabidopsis. PKS5 can interact with and phosphorylate SOS2 at Ser[294], promote the interaction between SOS2 and 14-3-3 proteins, and repress SOS2 activity. However, salt stress promotes an interaction between 14-3-3 proteins and PKS5, repressing its kinase activity and releasing inhibition of SOS2. We provide evidence that 14-3-3 proteins bind to $Ca^{2+}$, and that $Ca^{2+}$ modulates 14-3-3-dependent regulation of SOS2 and PKS5 kinase activity. Our results suggest that a salt-induced calcium signal is decoded by 14-3-3 and SOS3/SCaBP8 proteins, which selectively activate/inactivate the downstream protein kinases SOS2 and PKS5 to regulate $Na^+$ homeostasis by coordinately mediating plasma membrane $Na^+/H^+$ antiporter and $H^+$-ATPase activity.

[1] State Key Laboratory of Plant Physiology and Biochemistry, College of Biological Sciences, China Agricultural University, 100193 Beijing, China. [2] National Institute of Biological Sciences, Beijing, 7 Science Park Road, Zhongguancun Life Science Park, 102206 Beijing, China. [3] Collaborative Innovation Center of Crop Stress Biology, Henan Province, Institute of Plant Stress Biology, Henan University, 475001 Kaifeng, China. Correspondence and requests for materials should be addressed to Y.Y. (email: yangyongqing@cau.edu.cn) or to Y.G. (email: guoyan@cau.edu.cn)

Calcium, a universal secondary messenger, is an important regulator of many cellular activities in both plants and animals. Fluctuations in the concentration of cytosolic-free $Ca^{2+}$ ($[Ca^{2+}]_{cyt}$) triggered by internal or external stimuli are decoded by different $Ca^{2+}$ sensors, such as calmodulin (CaM)[1–3], $Ca^{2+}$-dependent protein kinases (CDPKs)[4,5], and SOS3-like $Ca^{2+}$-binding protein/calcineurin B-like protein (SCaBP/CBL)[6–11]. However, it is unclear how different calcium sensors decode a calcium signal and coordinately regulate the activity of various cellular targets to achieve a specific physiological response.

The salt overly sensitive (SOS) pathway, which is conserved in plants, regulates sodium ion homeostasis under salt stress[10,11]. The major components of the SOS pathway are the SOS3 and SCaBP8 calcium sensors, the SOS2 protein kinase, and the plasma membrane $Na^+/H^+$ antiporter SOS1 (PM $Na^+/H^+$ antiporter)[12–15]. Under salt stress, SOS3 and SCaBP8 perceive the salt-induced $Ca^{2+}$ signal and interact with SOS2, thereby recruiting it to the plasma membrane[14,16,17]. SOS2 then phosphorylates SOS1^Ser1138, which alleviates auto-inhibition of SOS1 by the C-terminal repressor domain, activating SOS1 and increasing $Na^+$ efflux[18–20]. Under normal growth conditions (in the absence of salt stress), SOS2 is phosphorylated at Ser294 and interacts with 14-3-3 proteins, which repress the kinase activity of SOS2[21]. Another protein, GI, also interacts with and represses SOS2 activity under normal growth conditions[22]. However, it is unknown which kinase phosphorylates SOS2^Ser294 and how 14-3-3 proteins are regulated to either bind or release SOS2 in the absence or presence of salt stress, respectively.

Activation of the SOS1 $Na^+/H^+$ antiporter under salt stress requires that SOS2 be activated and that a plasma membrane $H^+$-ATPase (PM $H^+$-ATPase)-generated proton gradient be established across the plasma membrane[23]. Activation of the PM $H^+$-ATPase is involved in phosphorylation/dephosphorylation processes and binding of 14-3-3 GF14-ω (14-3-3ω) protein to the PM $H^+$-ATPase AHA2 at Thr947 which relieves its auto-inhibition by the C-terminal domain[24–28]. SOS2-LIKE PROTEIN KINASE5 (PKS5) phosphorylates the PM $H^+$-ATPase AHA2 at Thr931 and inhibits its activity by reducing the binding of 14-3-3ω to AHA2^Thr947, which negatively regulates salt-alkaline tolerance of Arabidopsis[24]. Although it is clear that PM $H^+$-ATPase is activated under salt stress in plant to provide a driving force for the $Na^+/H^+$ antiporter, little is known about how these two transporters are coordinately regulated.

In this study, we show that PKS5 can interact with and phosphorylate SOS2. PKS5 can negatively regulate salt tolerance and provide evidence that PKS5 and SOS2 activity is regulated in a $Ca^{2+}$- dependent manner. We provide a model whereby 14-3-3 proteins act as a $Ca^{2+}$-dependent switch to coordinately regulate SOS2 and PKS5, thereby activating both the PM $Na^+/H^+$ antiporter and PM $H^+$-ATPase and mediating the plant's response to salt stress.

## Results

### PKS5 can interact with and phosphorylate SOS2 at Ser294.

Phosphorylation of SOS2^Ser294 is important for the regulation of SOS2 kinase activity. To identify the kinase responsible for phosphorylating SOS2^Ser294, we obtained Arabidopsis transgenic plants expressing Pro35S:6×Myc-SOS2 in the sos2-2 mutant background[17] and searched for potential SOS2 interactors by mass spectrometry analysis. Among the list of candidate-interacting proteins, we identified PKS5 (Supplementary Fig. 1a, Supplementary Data 1). We verified the interaction between PKS5 and SOS2 by yeast two-hybrid analysis (Fig. 1a). In addition, a bimolecular fluorescence complementation (BiFC) analysis in Nicotiana benthamiana revealed yellow fluorescent protein

(YFP) fluorescence signal in leaves transiently expressing $YFP^N$-PKS5 and $YFP^C$-SOS2, but not in those expressing $YFP^N$-AHA1 and $YFP^C$-SOS2 (Fig. 1b). Thus, PKS5 can interact with SOS2 in yeast and in plant cells.

We next examined whether PKS5 can phosphorylate SOS2 in vitro. In order to avoid interference caused by auto-phosphorylation of SOS2, we used a kinase-dead mutant variant[12], His-tagged SOS2^K40N, for the kinase assay, and the result revealed that SOS2^K40N was phosphorylated by PKS5 (Fig. 1c). We also performed an in vitro kinase assay to determine whether SOS2 phosphorylates PKS5, the result showed that SOS2 could not phosphorylate PKS5^K50N (a kinase-dead-type mutant) (Supplementary Fig. 1c). To identify the site of SOS2 that is phosphorylated by PKS5, we mapped the SOS2 phosphorylation region. For this, we purified three recombinant glutathione S-transferase (GST)-tagged SOS2-truncated proteins, namely GST-SOS2-KD (the N-terminal kinase domain), GST-SOS2-JD (the junction domain), and GST-SOS2-RD (the regulator domain)[21]. An in vitro phosphorylation assay showed that only the junction domain of SOS2 was phosphorylated by PKS5 (Fig. 1c). Ser294 is the only possible phosphorylation site in this peptide[21] and, as expected, mutation of this residue to Ala abolished the PKS5-mediated phosphorylation of SOS2^K40N and SOS2-JD. SOS2-JD^S294A displayed a protein shift compared to the wild-type SOS2-JD peptide (Fig. 1c, d), which suggests that the mutation may change the protein structure or charge. Furthermore, a mass spectrometry assay confirmed that PKS5 phosphorylated SOS2-Ser294 (Supplementary Fig. 1b). SCaBP1, a SOS3-like calcium-binding protein, can bind to $Ca^{2+}$ and activate PKS5 to repress PM $H^+$-ATPase activity in yeast[24]. We therefore tested whether $Ca^{2+}$ and SCaBP1 play a role in the phosphorylation of SOS2 by PKS5. In vitro phosphorylation assay showed that $Ca^{2+}$ and SCaBP1 had no obvious effect on the phosphorylation of SOS2-JD by PKS5 (Supplementary Fig. 1d). Overall, these results demonstrate that PKS5 can interact with and phosphorylate SOS2 at Ser294.

### PKS5 enhances the interaction between SOS2 and 14-3-3s.

As PKS5 can phosphorylate SOS2^Ser294, we wondered whether PKS5 plays a role in the 14-3-3-mediated repression of SOS2 kinase activity. We first examined whether PKS5 affects the interaction between 14-3-3 proteins and SOS2 in vitro. For this, GST-tagged SOS2-JD, GST-tagged SOS2-JD^S294A, His-tagged PKS5, and His-tagged 14-3-3λ proteins purified from Escherichia coli were used for an in vitro pull-down assay. We observed that SOS2-JD interacts with 14-3-3λ, and that this interaction was enhanced by the addition of PKS5 particularly in the presence of kinase buffer containing ATP (Fig. 2a). However, the enhancement was reduced when the SOS2^S294A mutant was used (Fig. 2a). To determine the effect of PKS5 on the 14-3-3/SOS2 interaction, we crossed Col-0 expressing Pro35S:6×Myc-SOS2 with the PKS5 loss-of-function mutant pks5-1[24], and the gain-of-function mutants, pks5-3 and, pks5-4, the latter of which were isolated in the Col er105 background (subsequently referred to as BigM) and have higher PKS5 kinase activity[29]. Using these crossed lines, Myc-SOS2 was immunoprecipitated with anti-C-Myc agarose and 14-3-3 proteins were detected with anti-14-3-3 antibody in the co-immunoprecipitated complex. Higher 14-3-3 protein signal resulted from SOS2 immunoprecipitation in the pks5-3 and pks5-4 backgrounds than in the BigM control; however, less immunoprecipitated 14-3-3 protein signal was observed in the pks5-1 background than in Col-0 (Fig. 2b). Split-luciferase complementation (LUC) imaging assays in N. benthamiana also indicated that the presence of PKS5 enhanced the interaction between SOS2 and 14-3-3λ, and that the enhancement was

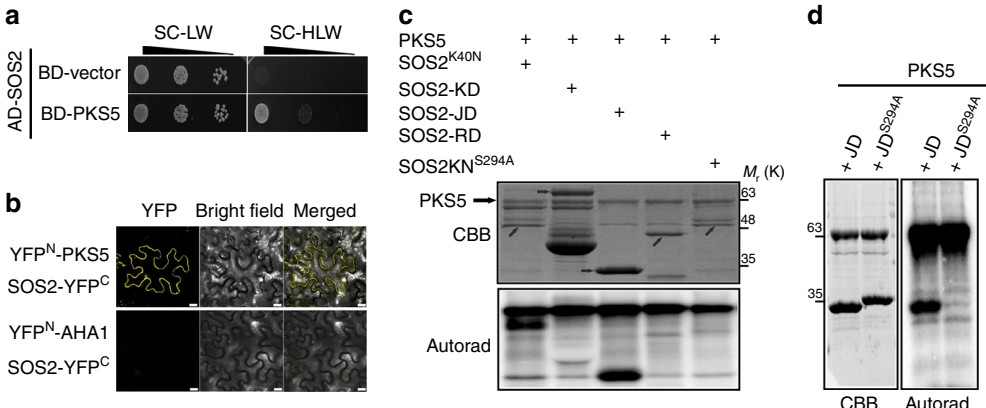

**Fig. 1** PKS5 interacts with SOS2 and phosphorylates SOS2Ser294. **a** Yeast two-hybrid assay showing that SOS2 interacts with PKS5. No interaction was observed when using an empty-vector control (BD-vector). **b** BiFC assay of the SOS2/PKS5 interaction in *N. benthamiana*. Plasmids containing *YFP[N]-PKS5* and *SOS2-YFP[C]* or *YFP[N]-AHA1* and *SOS2-YFP[C]* were co-transformed into *N. benthamiana* leaves. Two days post-infiltration, the YFP fluorescence was detected by confocal microscopy. Scale bars, 50 μm. **c** In vitro kinase assay showing that PKS5 phosphorylates SOS2Ser294. The recombinant proteins were incubated in kinase buffer at 30 °C for 30 min and separated by SDS–PAGE. Arrow indicates the target protein. **d** In vitro kinase assay showing that PKS5 phosphorylates SOS2-JD but not SOS2-JDS294A. JD, SOS2-JD; JDS294A, SOS2-JDS294A. CBB Coomassie brilliant blue, Autorad autoradiograph. Source data of **c** and **d** are provided in the Source Data file

greater when the constitutively active kinases PKS5-3 and PKS5-4 were used (Fig. 2c, d). However, the SOS2S294A mutant abolished the PKS5-enhanced interaction between SOS2 and 14-3-3λ, and co-expressed PKS5K50N had no significant effect on this interaction (Supplementary Fig. 2b, c). These results suggest that PKS5 enhances the interaction between SOS2 and 14-3-3λ through phosphorylating the SOS2 Ser294. We also tested whether SCaBP1 has effect on the PKS5-mediated interaction between SOS2 and 14-3-3λ, and LUC imaging assays showed that SCaBP1 had no obvious effect on this interaction (Supplementary Fig. 2d, e).

14-3-3λ and κ bind to and regulate SOS2 kinase activity[21]. We thus analyzed the effect of PKS5 on SOS2 kinase activity *in planta*. We first conducted comparative phosphorylation assays using His-tagged SCaBP8 and MBP as substrates for SOS2, and SOS2 immunoprecipitated from transgenic plants expressing Myc-SOS2 in BigM, *pks5-3*, *pks5-4*, Col-0, and *pks5-1* backgrounds. NaCl treatment is known to enhance SOS2 kinase activity[17]. The kinase activity of Myc-SOS2 in BigM, *pks5-3*, and *pks5-4* was low before NaCl treatment, and NaCl-induced activation of SOS2 was repressed in *pks5-3* and *pks5-4* (Fig. 2e). The base-level kinase activity of SOS2 was increased in *pks5-1*, and was not further enhanced following NaCl treatment (Fig. 2f). SCaBP8Ser237 is specifically phosphorylated by SOS2[17], and thus we used anti-phospho-SCaBP8S237 (anti-P-SC8)[17] to examine SCaBP8 phosphorylation as a measure of SOS2 kinase activity. Total proteins were extracted from transgenic plants expressing *35S:6×Myc-SCaBP8* in the *pks5* and wild type backgrounds with or without NaCl treatment, and SCaBP8 protein and phosphorylation levels were determined by immunoblot analysis using anti-C-Myc and anti-P-SC8 antibodies, respectively. Consistently, we found that SCaBP8 phosphorylation in *pks5-3* and *pks5-4* was lower than in the wild type following NaCl treatment (Fig. 2g), and was higher in *pks5-1* than in the wild type prior to NaCl treatment (Fig. 2h). These results suggest that PKS5 negatively regulates SOS2 kinase activity by phosphorylating SOS2.

**PKS5 inhibits the SOS pathway in yeast through 14-3-3s**. To determine the effect of PKS5-mediated SOS2Ser294 phosphorylation by PKS5 on the SOS pathway, we performed an experiment in a yeast system named as Sos Recruitment System (SRS) in which the SOS pathway was reconstituted via coexpression of SOS1 and constitutively activated SOS2T168D/Δ308 in yeast mutants lacking

endogenous Na+ transporters[20]. We mutated Ser294 of SOS2 to Ala or Asp to generate the phospho-dead or phosphor-mimic status of the protein, respectively. Subsequently, we expressed *SOS2T168D/Δ308/S294A*, *SOS2T168D/Δ308/S294D*, and *SOS2T168D/Δ308* constructs with or without 14-3-3λ in yeast strain *Saccharomyces cerevisiae* JP837, and assessed the ability of the resulting transformants to grow on Arg phosphate (AP) medium containing NaCl. 14-3-3λ expression did not affect yeast growth on AP medium containing 25 mM NaCl, but limited growth on medium containing 50 mM NaCl (Supplementary Fig. 2h). There was no growth difference on salt medium when *SOS2T168D/Δ308/S294A* was expressed compared to *SOS2T168D/Δ308* in the presence or absence of 14-3-3λ (Supplementary Fig. 2f). By contrast, expression of the Ser294-to-Asp mutant form (*SOS2T168D/Δ308/S294D*), which enhances the 14-3-3λ/SOS2 interaction and represses SOS2 kinase activity[21], strongly repressed yeast growth on AP medium that contained only 15 mM NaCl (Supplementary Fig. 2g), and coexpression of *14-3-3λ* further increased this repression but had little effect on yeast expressing *SOS2T168D/Δ308* (Supplementary Fig. 2g).

It is possible that 14-3-3λ homologs in yeast repress SOS2T168D/Δ308/S294D kinase activity. However, the addition of PKS5 failed to repress SOS2 activity in yeast (Supplementary Fig. 2h). Furthermore, the efficient function of PKS5 may require the SOS3-like Ca2+-binding protein SCaBP1[24]. As a control, 14-3-3ω did not obviously affect SOS2 activity in the yeast system (Supplementary Fig. 2h). When *SOS2T168D/Δ308* was coexpressed with *14-3-3λ*, *PKS5-3/PKS5-4* substantially repressed yeast growth, consistent with PKS5 phosphorylation and repression of SOS2T168D/Δ308 (Fig. 2i), whereas the coexpression of *14-3-3λ* and *SOS2T168D/Δ308/S294D* resulted in no further growth inhibition in the presence of PKS5-3/PKS5-4 (Fig. 2j). Collectively, these results further suggest that PKS5-dependent phosphorylation of SOS2Ser294 represses SOS2 kinase activity. In yeast this appears to be dependent on the co-expression of 14-3-3λ-dependent manner.

**PKS5 negatively regulates salt tolerance in Arabidopsis**. Given our evidence that PKS5 can phosphorylate SOS2 and regulate SOS2 kinase activity, we next determined whether PKS5 affects salt tolerance in Arabidopsis. Following the transfer of 7-day-old seedlings to medium with 75 or 100 mM NaCl, *pks5-3* and *pks5-4* with enhanced PKS5 activity showed significantly reduced root

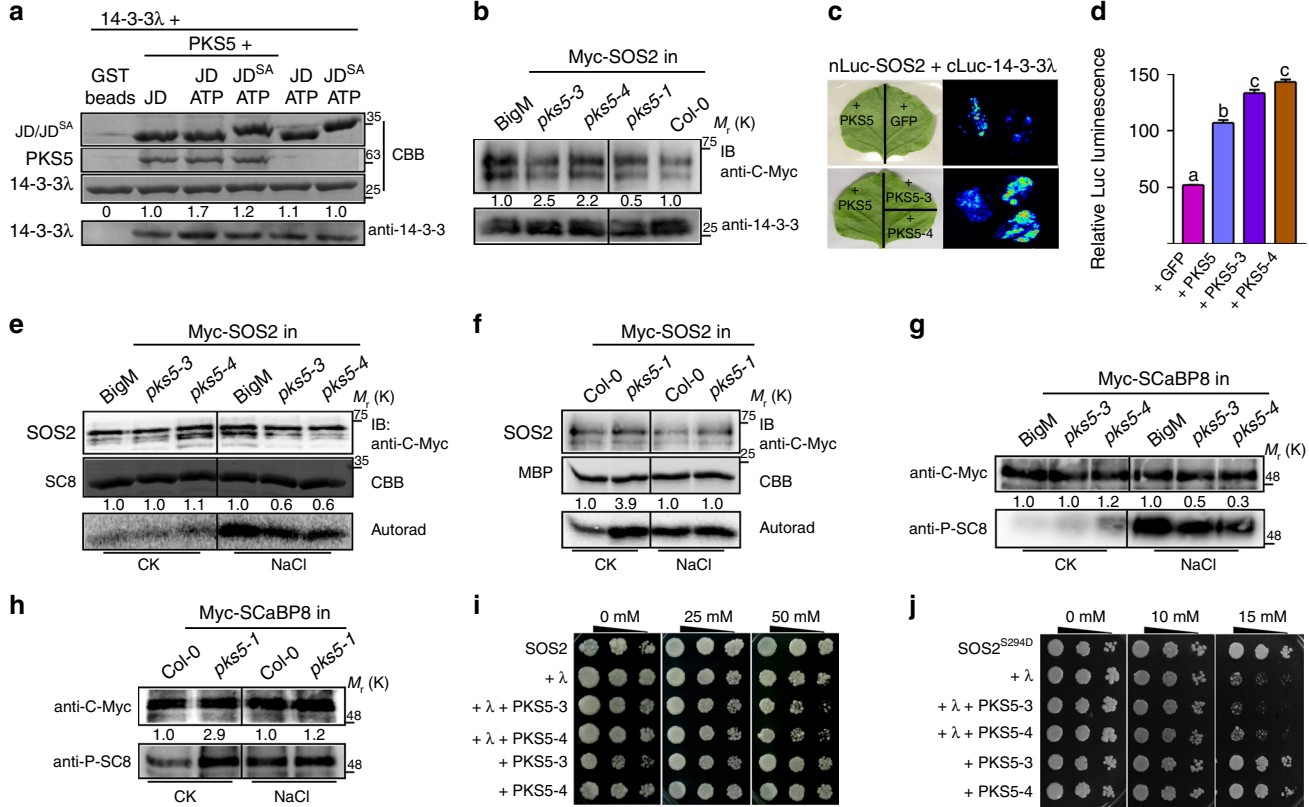

**Fig. 2** PKS5 enhances the interaction between SOS2 and 14-3-3 proteins. **a** In vitro pull-down assay showing that PKS5 enhances the interaction between SOS2-JD and 14-3-3λ following incubation with SOS2-JD in kinase buffer containing ATP. JD, SOS2-JD; JD$^{SA}$, SOS2-JD$^{S294A}$. **b** In vivo Co-immunoprecipitation assay showing that PKS5 enhances the interaction between SOS2 and 14-3-3 proteins. Myc-SOS2 was extracted with anti-C-Myc agarose from 10-day-old seedlings expressing *Pro35S:6×Myc-SOS2* in BigM/*pks5-3*/*pks5-4*/Col-0/*pks5-1* backgrounds and analyzed by immuno blotting with anti-C-Myc and anti-14-3-3 antibodies. **c** Luciferase complementation imaging assay showing that co-expression of PKS5/PKS5-3/PKS5-4 in *N. benthamiana* enhanced the interaction between SOS2 and 14-3-3λ. **d** Relative fluorescence analysis of (**c**) by ImageJ. Error bars represent SD; $p ≤ 0.05$, Student's *t*-test; $n = 3$; significant differences are indicated by different lowercase letters. **e**, **f** In vitro kinase assay showing that PKS5 negatively regulates SOS2 activity. Myc-SOS2 was extracted using anti-C-Myc agarose from 10-day-old transgenic seedlings treated with or without 100 mM NaCl for 12 h. His-SCaBP8 (SC8) and MBP were used as the substrates and anti-C-Myc antibody was used for immunoblot analysis. **g**, **h** In vivo Co-immunoprecipitation assay showing that PKS5 negatively regulates the phosphorylation status of SCaBP8. Myc-SCaBP8 was extracted from 10-day-old seedlings expressing *Pro35S:6×Myc-SCaBP8* in the BigM/*pks5-3*/*pks5-4*/Col-0/*pks5-1* backgrounds treated or not with 100 mM NaCl for 12 h and analyzed by immunoblotting with anti-C-Myc and anti-phospho-SCaBP8Ser237 (anti-P-SC8) antibodies. **i**, **j** SOS recruitment system assays showing that PKS5 inhibits the SOS pathway by phosphorylating SOS2$^{Ser294}$ and enhances the interaction between SOS2 and 14-3-3λ. Yeast cells expressing Arabidopsis SOS1 were co-transformed with the indicated plasmids, two positive clones with 3.5 μL of serial five-fold dilutions were grown on AP medium with different concentrations of NaCl to analyze the salt tolerance of yeast strains expressing SOS2, SOS2T168D/Δ308; SOS2$^{S294D}$, SOS2T168D/Δ308/S294D. IB immunoblot. CBB Coomassie Brilliant Blue, Autorad autoradiograph. Source data of **a**, **b**, and **e**–**h** are provided in the Source Data file

growth compared to the wild type, and the reduction was greater in the presence of 100 mM than 75 mM NaCl (Fig. 3a, b). However, the root growth of *pks5-1* was greater than that of Col-0 when seedlings were transferred to medium containing 125 mM NaCl (Supplementary Fig. 3a, b). Furthermore, using Non-invasive Micro-test Technology (NMT), we measured Na$^+$ flux at the meristem zone of 7-day-old BigM, *pks5-3*, and *pks5-4* seedlings following a 12-h treatment with 100 mM NaCl. Salt-stressed roots exhibited a pronounced net Na$^+$ efflux, however, efflux was significantly lower in *pks5-3* and *pks5-4* than in BigM (Fig. 3c, d).

To further dissect the function of PKS5-mediated SOS2 phosphorylation, we expressed Myc-SOS2 in *pks5-4* (S2) and Myc-SOS2$^{S294A}$ in *pks5-4* (S2SA) plants by crossing transgenic Arabidopsis plants expressing *35SPro:6×Myc-SOS2* or *35SPro:6×-Myc-SOS2$^{S294A}$* with *pks5-4*. We found that the SOS2 and SOS2$^{S294A}$ protein levels were similar in the resulting crossed lines (Supplementary Fig. 3e). Overexpression of SOS2 did not rescue the *pks5-4* salt-sensitive phenotype; however, the

salt-sensitive phenotype of *pks5-4* was rescued by the expression of SOS2$^{S294A}$ (Fig. 3e, f), with different independent transgenic lines showing a similar phenotype (Supplementary Fig. 3c, d). We next measured Na$^+$ flux in the crossed lines, and found that SOS2$^{S294A}$ expression rescued the faulty Na$^+$ flux of *pks5-4*, whereas expression of SOS2 did not (Fig. 3g, h).

Thus, PKS5 negatively regulates salt tolerance in Arabidopsis in a manner dependent on SOS2$^{Ser294}$ phosphorylation.

**Salt stress enhances the interaction between 14-3-3 and PKS5.** SOS2 is activated by salt stress[17]. As we found that PKS5 represses SOS2 activity and enhances the 14-3-3–SOS2 interaction, and that PKS5-mediated suppression of SOS2 in yeast is 14-3-3 dependent, we hypothesized that PKS5 represses SOS2 activity under normal growth conditions and that salt stress relieves this repression. Thus, we examined whether salt stress affects the kinase activity of PKS5 *in planta* by generating

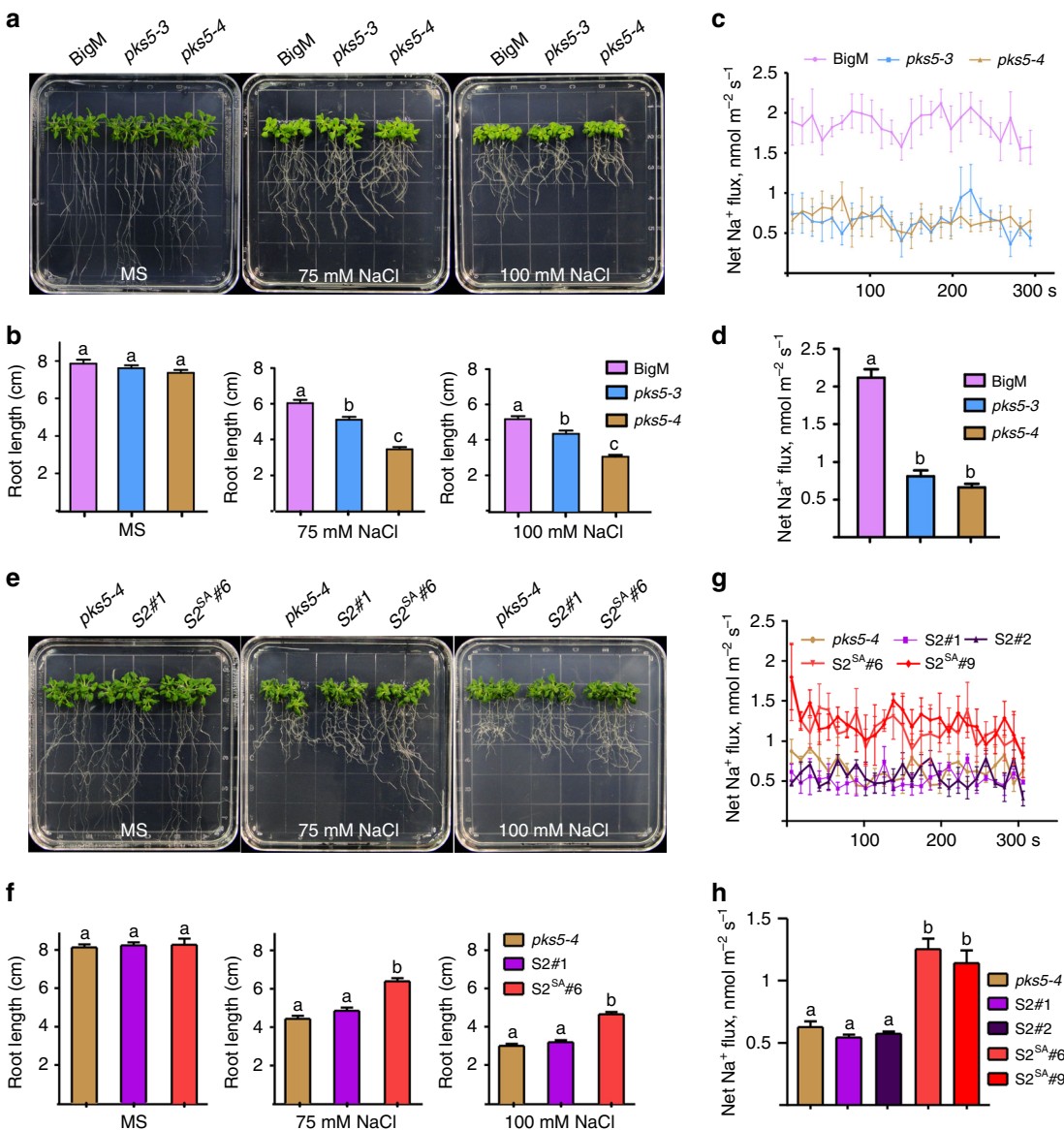

**Fig. 3** SOS2[Ser294] mutation rescues the salt-sensitive phenotype of *pks5-4*. **a** Salt sensitive analysis of BigM, *pks5-3*, and *pks5-4*. Five-day-old seedlings were transferred from MS to medium containing 75 or 100 mM NaCl. Photographs were taken 10 days after transfer. **b** Root length analysis of seedlings in (**a**). Error bars represent SD; $p \leq 0.05$, Student's *t*-test; $n = 12$. **c** Net Na[+] flux analysis at the meristem zone of BigM, *pks5-3*, and *pks5-4* by NMT. Five-day-old seedlings were pretreated with 100 mM NaCl for 24 h. Error bars represent SD; $p \leq 0.05$, Student's *t*-test; $n = 6$. **d** Calculated net Na[+] fluxes from (**c**). **e** Salt sensitive analysis of *pks5-4* and transgenic plants expressing Myc-SOS2 or Myc-SOS2[S294A] in the *pks5-4* background. S2, *Pro35S:6×Myc-SOS2* in *pks5-4*; S2[SA], *Pro35S:6×Myc-SOS2S294A* in *pks5-4*. Photographs were taken 10 days after transfer. **f** Root length analysis of seedlings in (**e**). Error bars represent SD; $p \leq 0.05$, Student's *t*-test; $n = 12$. **g** Net Na[+] flux analysis at the meristem zone of *pks5-4* and different transgenic plants by NMT. Error bars represent SD; $p \leq 0.05$, Student's *t*-test; $n = 6$. **h** Calculated net Na[+] fluxes from (**g**). For **b**, **d**, **f**, and **h**, significant differences are indicated by different lowercase letters. Source data of **b**–**d** and **f**–**h** are provided in the Source Data file

transgenic Arabidopsis plants overexpressing *Myc-PKS5* in Col-0, with expression driven by the 35S promoter. Using anti-C-Myc agarose, Myc-PKS5 protein was immunoprecipitated from plants treated with 100 mM NaCl for various periods and subjected to kinase assay, which showed that NaCl treatment repressed the kinase activity of PKS5 (Fig. 4a).

We then sought to identify which protein(s) repress PKS5 kinase activity under salt stress. Salt stress reduces the interaction between SOS2 and 14-3-3 proteins[21], and PKS5 and SOS2 belong to the same sucrose non-fermenting-1-related protein kinase-3 (SnRK3) protein family. We thus hypothesized that 14-3-3 proteins repress the kinase activity of PKS5 under salt stress.

To test this hypothesis, we analyzed the interaction between 14-3-3λ and PKS5 in a yeast two-hybrid assay. We found that 14-3-3λ interacted with the N-terminal kinase domain of PKS5 (PKS5N, amino acids 1–280) but not the junction domain (PKS5JD, amino acids 281–306) or C-terminal regulatory domain (PKS5C, amino acids 307–435) (Fig. 4b). Co-immunoprecipitation assays confirmed that PKS5 interacted with 14-3-3λ, and that this interaction was induced by NaCl treatment (Fig. 4c).

A LUC imaging assay confirmed that PKS5 interacted with 14-3-3λ and that NaCl treatment enhanced this interaction (Fig. 4d, e), thus further supporting our conclusion that salt stress enhances the interaction between 14-3-3 proteins and PKS5.

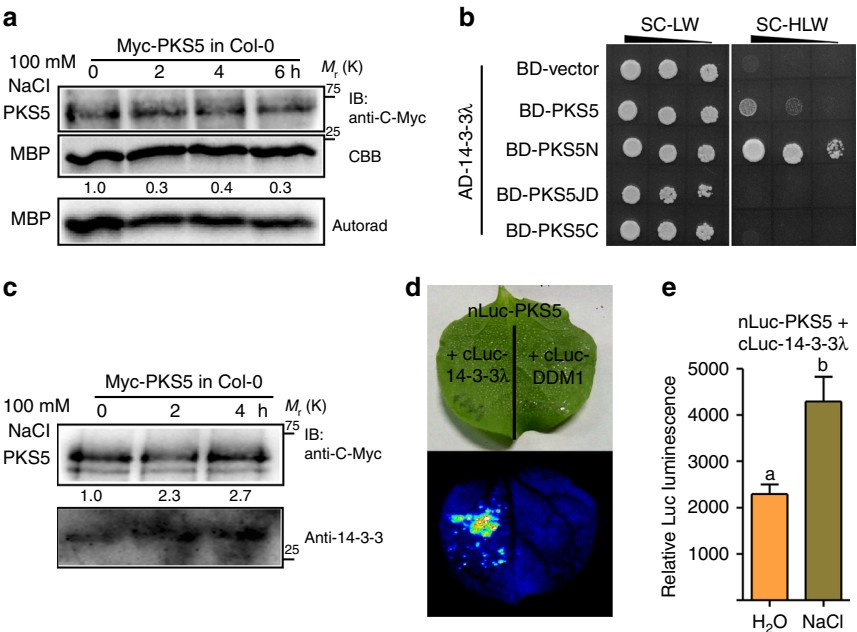

**Fig. 4** Salt stress enhances the interactions between 14-3-3 proteins and PKS5. **a** In vitro kinase assay shows that salt stress decreases PKS5 activity. Myc-PKS5 was extracted from 10-day-old Col-0 seedlings transgenically expressing *Pro35S:6×Myc-PKS5* treated with 100 mM NaCl for the indicated periods. MBP was used as the substrate. **b** Yeast two-hybrid assay showing that 14-3-3λ interacts with the kinase domain of PKS5. PKS5N, kinase domain (amino acids 1–280); PKS5JD, junction domain (amino acids 281–306); PKS5C, C-terminal regulatory domain (amino acids 307–435). **c** In vivo Co-immunoprecipitation assay showing that salt stress enhances the interaction between PKS5 and 14-3-3 proteins. Myc-PKS5 was extracted from Col-0 expressing *Pro35S:6×Myc-PKS5* treated with 100 mM NaCl for the indicated periods, and then analyzed by immunoblotting with anti-C-Myc and anti-14-3-3 antibodies. **d** PKS5 interacts with 14-3-3λ in a Luciferase complementation assay. DDM1 was used as a negative control. **e** Luciferase complementation imaging assay showing that treatment with 200 mM NaCl enhances the interaction between PKS5 and 14-3-3λ. Error bars represent SD; $p \leq 0.05$, Student's *t*-test; $n = 6$; significant difference is indicated by different lowercase letters. CBB Coomassie Brilliant Blue, Autorad autoradiograph, IB immunoblot. Source data of **a**, **c**, and **e** are provided in the Source Data file

**14-3-3 proteins repress the kinase activity of PKS5.** The finding that salt stress reduces the kinase activity of PKS5 and increases the interaction between 14-3-3λ and PKS5 led us to hypothesize that 14-3-3λ may directly repress PKS5 activity via protein–protein interaction under salt stress. To test this hypothesis, we performed an in vitro kinase assay using recombinant His-tagged PKS5 and GST-tagged 14-3-3λ, 14-3-3κ, 14-3-3ω, and SCaBP1 (a substrate for PKS5) from Arabidopsis following heterologous expression and purification from *E. coli*. We found that both 14-3-3λ and 14-3-3κ repressed PKS5 transphosphorylation activity against SCaBP1 and also repressed PKS5 autophosphorylation. 14-3-3ω, included as a control, did not markedly affect SCaBP1 phosphorylation (Fig. 5a) and the inhibition of SCaBP1 phosphorylation by 14-3-3λ was dose dependent. To further investigate the effect of 14-3-3 proteins on PKS5 activity in vivo, we performed an IP-kinase assay. We generated transgenic Arabidopsis plants overexpressing *Myc-PKS5* in the *14-3-3λκ* double mutant background. Ten-day-old transgenic seedlings were treated with 100 mM NaCl for 2 or 4 h, and then PKS5 was immunoprecipitated by anti-C-Myc agarose. Consistent with previous results (Fig. 4a), the phosphorylation of MBP was repressed in Col-0 by salt stress (Fig. 5b); however, no significant difference in PKS5 kinase activity was observed after NaCl treatment between the *14-3-3λκ* double mutant and Col-0 (Fig. 5c). These results suggest that 14-3-3 proteins repress PKS5 activity under salt stress.

Previous studies suggest that PKS5 negatively regulates PM H$^+$-ATPase activity, and that the PKS5 loss-of-function mutant *pks5-1* is tolerant to high external pH[24]. As 14-3-3λ and κ both repress PKS5 activity, we examined the growth of *14-3-3λκ*

double mutant seedlings in response to high external pH. On pH 6.5 and 6.7 MS medium, root elongation was inhibited to a greater extent in the *14-3-3λκ* double mutant than in Col-0 (Supplementary Fig. 4a, b).

Furthermore, we measured PM H$^+$-ATPase activity using plasma membrane-enriched vesicles[29], which were isolated from 30-day-old Col-0 or *14-3-3λκ* double mutant plants that had been subjected to NaCl treatment for 3 days. We found that the NaCl-induced increase in PM H$^+$-ATPase activity was significantly lower in the *14-3-3λκ* double mutant than in Col-0 (Supplementary Fig. 4c, d). NMT measurement also indicated that the H$^+$ efflux of *14-3-3λκ* double mutants was lower than that of Col-0 (Supplementary Fig. 4e, f).

Phosphorylation of AHA2 C-terminal Thr$^{947}$ generates a binding site that promotes interaction with the 14-3-3ω protein and activation of the PM H$^+$-ATPase[30]. We examined if 14-3-3λ and 14-3-3κ also interact with AHA2. Yeast two-hybrid analysis showed that when Thr$^{947}$ was mutated to Asp, 14-3-3ω interacted with AHA2 C100$^{T947D}$, whereas no interaction was detected between 14-3-3λ/κ and AHA2 C100$^{T947D}$ (Supplementary Fig. 4i). LUC imaging assays also showed that AHA2 C100$^{T947D}$ (947D) strongly interacted with 14-3-3ω but not 14-3-3λ, 14-3-3κ, or the negative control DDM1 in transgenic *N. benthamiana* plants (Supplementary Fig. 4g, h).

In addition, we measured the effect of GST-14-3-3 proteins on PM H$^+$-ATPase activity using plasma membrane vesicles isolated from Col-0 after 3 days of NaCl treatment. Consistent with previous results, 14-3-3ω protein significantly increased PM H$^+$-ATPase activity, whereas 14-3-3λ and 14-3-3κ proteins had no effect on PM H$^+$-ATPase activity, similar to the

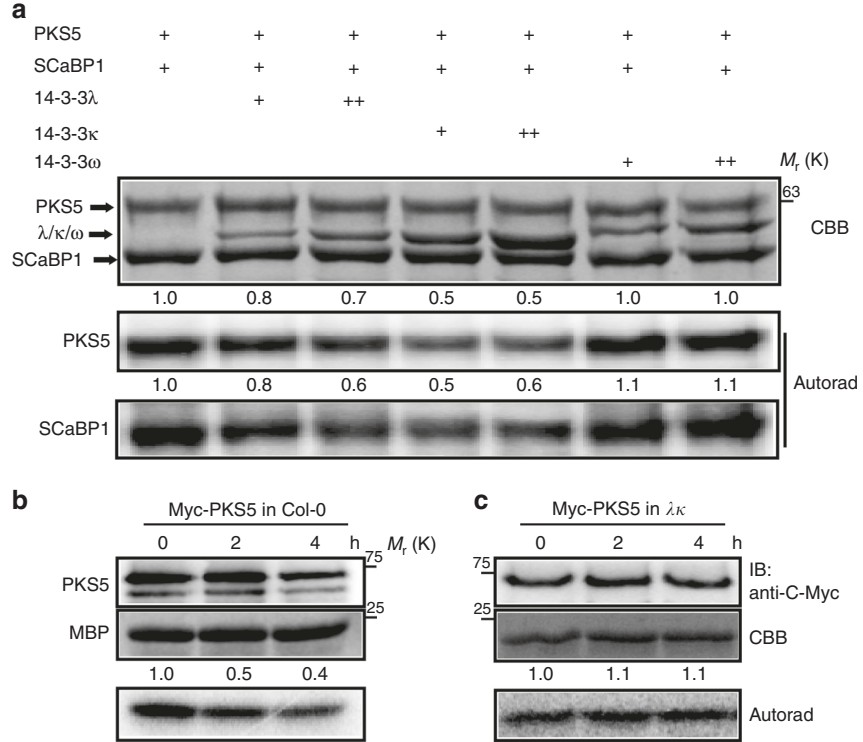

**Fig. 5** 14-3-3 proteins repress the kinase activity of PKS5. **a** In vitro kinase assay showing that 14-3-3λ and κ proteins inhibit PKS5 activity in a dose-dependent manner. Recombinant His-tagged PKS5 was incubated with GST-tagged SCaBP1 with or without GST-tagged 14-3-3 proteins. SCaBP1 was used as the substrate and 14-3-3ω served as a negative control. **b**, **c** Kinase assays of PKS5 in Col-0 and *14-3-3λκ* double mutant backgrounds under salt stress. Myc-PKS5 was purified from *Pro35S:6×Myc-PKS5* transgenic plants treated with 100 mM NaCl for the indicated periods. MBP was used as the substrate. CBB Coomassie Brilliant Blue, Autorad autoradiograph, IB immunoblot. Source data are provided as a Source Data file

GST-tagged protein, which was used as a negative control (Supplementary Fig. 4j). These results suggest that 14-3-3λ and 14-3-3κ regulate PM H$^+$-ATPase activity by repressing PKS5 activity.

**14-3-3s function in PKS5-mediated alkaline stress response.** Given that PKS5 negatively regulates PM H$^+$-ATPase activity, the loss-of-function *pks5-1* mutant showed an alkaline tolerance phenotype, whereas the gain-of-function *pks5-3* and *pks-4* mutants showed an alkaline sensitive phenotype[29]. To test whether 14-3-3λ and κ are required for plant responses to alkaline stress, we crossed Col-0 expressing *Pro35S:Flag-HA-14-3-3λ*[21] with *pks5-1* or *pks5-4* mutant plants to overexpress 14-3-3λ in the *pks5-1* (OE-1 and OE-2) and *pks5-4* (OE-3 and OE-4) backgrounds. Two independent lines were subsequently used in an alkaline tolerance assay. We found that overexpression of 14-3-3λ in *pks5-4* repressed the alkaline-sensitive phenotype of the mutant as well as PM H$^+$-ATPase activity (Fig. 6a–c); however, the alkaline tolerance phenotype and high PM H$^+$-ATPase activity of *pks5-1* were not repressed (Fig. 6d–f). Furthermore, overexpression of 14-3-3λ in *pks5-1* did not affect PM H$^+$-ATPase activity (Fig. 6e, f), whereas overexpression of 14-3-3λ in *pks5-4* partially rescued PM H$^+$-ATPase activity (Fig. 6b, c). These results further indicate that 14-3-3λ and κ regulate the alkaline stress response in Arabidopsis at least partly by repressing of PKS5 kinase activity.

**14-3-3 binds to calcium that regulates SOS2/PKS5 activity.** A salt stress-induced increase in cytosolic Ca$^{2+}$ concentration decreases the interaction between 14-3-3 and SOS2 and enhances the interaction between 14-3-3 and PKS5. It has been reported that 14-3-3λ and κ contain a conserved Ca$^{2+}$-binding EF-hand

motif[31]. Thus, we hypothesized that 14-3-3λ and κ play a role in regulating PKS5 and SOS2 kinase activity through calcium signals. We first tested whether 14-3-3 proteins can bind Ca$^{2+}$ in a microscale thermophoresis (MST) assay, and found that, in contrast to the negative control (GST-tagged protein), GST-14-3-3λ and GST-14-3-3κ bound Ca$^{2+}$ directly, the calcium-binding protein CDPK6 served as a positive control (Fig. 7a). In vitro kinase assays suggest Ca$^{2+}$ released SOS2 kinase activity from 14-3-3-mediated inhibition (Fig. 7b; Supplementary Fig. 5e). By contrast, the addition of Ca$^{2+}$ repressed PKS5 kinase activity (Fig. 7c; Supplementary Fig. 5f). The addition of the same amount of Ca$^{2+}$ did not directly affect the activities of PKS5 and SOS2 (Supplementary Fig. 5g, h). Furthermore, we found that the salt-enhanced 14-3-3/PKS5 and SOS3/SOS2 interactions were rescued by the addition of LaCl$_3$, a calcium channel blocker (Supplementary Fig. 5a–d) and that the effects of Ca$^{2+}$ on the repression of PKS5 and activation of SOS2 activities were rescued by EGTA (Supplementary Fig. 5e, f). These results suggest that the salt-induced calcium signal may positively regulate the 14-3-3/PKS5 interaction and regulates the activity of PKS5 and SOS2 kinases.

Previous studies demonstrated that 14-3-3ω binds to calcium at its C-terminus via an EF-hand-like motif[32]. We analyzed eight 14-3-3 homologs in Arabidopsis that contain a putative EF-hand, and found that 14-3-3λ$^{G216}$ is the only conserved amino acid in 8 of 13 14-3-3 proteins (Supplementary Fig. 6a). We mutated this residue to Ala and purified the recombinant protein from *E. coli*, and following a test of Ca$^{2+}$-binding activity by MST assays, found that 14-3-3λ$^{G216A}$ did not bind to Ca$^{2+}$ (Fig. 7d). Yeast two-hybrid and LUC imaging assays showed that the 14-3-3λ$^{G216A}$ mutation had no obvious effect on its ability to interact with PKS5 and SOS2, with protein interaction levels observed for 14-3-3λ$^{G216A}$ being comparable to those of wild-type 14-3-3λ

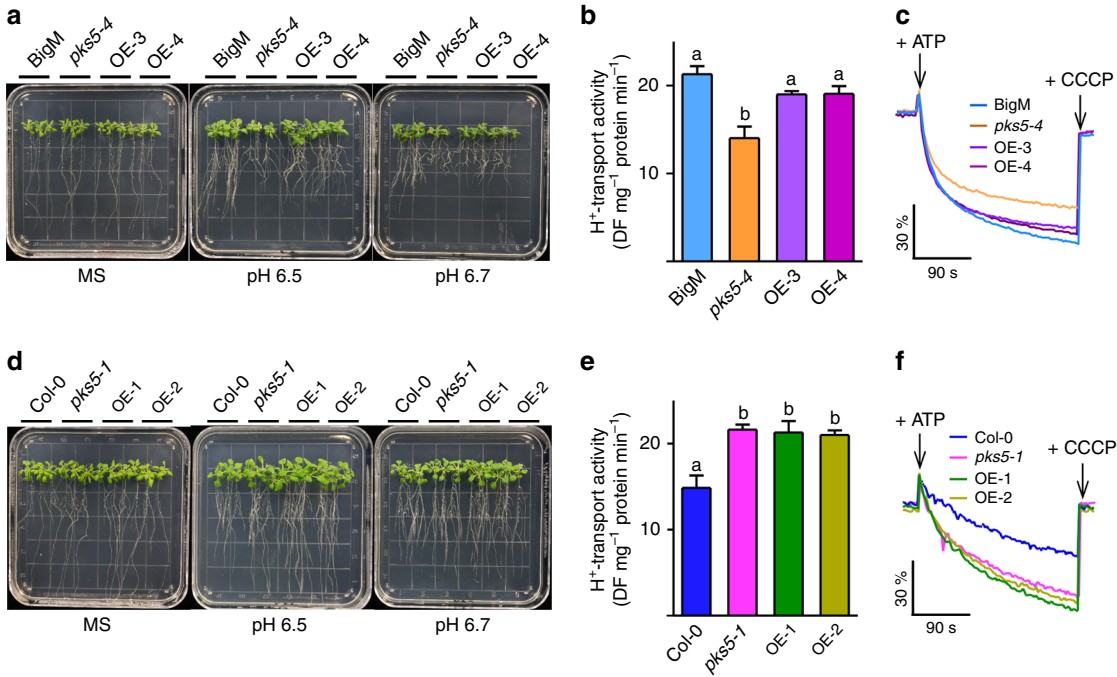

**Fig. 6** Overexpression of 14-3-3 rescues the phenotypes of *pks5-4*. **a** Alkaline sensitive analysis of BigM, *pks5-4*, and two independent transgenic lines (OE-3 and OE-4) expressing *Flag-HA-14-3-3λ* driven by the 35S promoter in the *pks5-4* background. Seven-day-old seedlings were transferred to pH 6.5 or 6.7 MS medium adjusted using NaHCO₃. Photographs were taken 10 days after seedling transfer. **b** Comparison of PM H⁺-ATPase activity in BigM, *pks5-4*, OE-3, and OE-4. Plasma membrane vesicles were purified from 5-week-old seedlings grown in soil treated with 250 mM NaCl for 3 days. Error bars represent SD; $p \leq 0.05$, Student's *t*-test; $n = 3$; significant differences are indicated by different lowercase letters. **c** Timely varying curves of PM H⁺-ATPase activity in BigM, *pks5-4*, OE-3, and OE-4. **d** Alkaline tolerance analysis of Col-0, *pks5-1*, and two independent transgenic lines (OE-1 and OE-2) expressing *Flag-HA-14-3-3λ* driven by the 35S promoter in the *pks5-1* background. Photographs were taken 10 days after seedling transfer. **e** Comparison of PM H⁺-ATPase activity in Col-0, *pks5-1*, OE-1, and OE-2. Error bars represent SD; $p \leq 0.05$, Student's *t*-test; $n = 3$; significant differences are indicated by different lowercase letters. **f** Timely varying curves of PM H⁺-ATPase activity in Col-0, *pks5-1*, OE-1, and OE-2. Source data of **b**, **c**, **e**, and **f** are provided in the Source Data file

(Supplementary Fig. 6b, d). And transient gene expression levels of *14-3-3λ*, *14-3-3λ*[G216A], *PKS5*, and *SOS2* in the LUC-imaging assay were similar (Supplementary Fig. 6c). Furthermore, we tested if calcium affects the ability of 14-3-3λ[G216A] to repress SOS2 and PKS5 kinase activity. In vitro kinase assays using 14-3-3λ[G216A] showed that the addition of Ca²⁺ did not change the 14-3-3λ[G216A]-mediated repression of PKS5 and SOS2 activities (Fig. 7e, f), although weak repressive activity towards SOS2 and PKS5 was observed when using 14-3-3λ[G216A] (Supplementary Fig. 6e). Taken together, our results suggest that different calcium concentrations are involved in regulating 14-3-3-mediated SOS2 and PKS5 activation and repression.

## Discussion

The calcium signal-activated SOS pathway is a conserved and important pathway that mediates plant sodium ion homeostasis under salt stress. According to previous studies and the results presented here, we propose a working model of how 14-3-3 proteins decode a calcium signal to enhance the salt tolerance of plants (Fig. 7g). Under normal growth conditions, PKS5 may phosphorylate SOS2[Ser294], and promote the 14-3-3/SOS2 interaction that in turn could inhibit SOS2 kinase activity and limit SOS1 Na⁺/H⁺ antiporter activity to basal levels. Furthermore, PKS5 can phosphorylate PM H⁺-ATPases, thus limiting their activity, which is required to provide the driving force for the Na⁺/H⁺ antiporter activity of SOS1. It is important that, under 'none-stressed' growth conditions, the proton ATPase maintains membrane potential and solute/nutrient uptake.

Therefore, precise regulation of PM H⁺-ATPase activity is important for plant growth and stress response. Under salt stress, the salt-induced calcium signal is decoded by various Ca²⁺-binding proteins, including 14-3-3s, SOS3, and SCaBP8, which may result in increased repression of PKS5 activity that reduces SOS2[Ser294] phosphorylation and in turn, relieves the repression of SOS2 activity by 14-3-3 proteins. In addition, a previous study reported that the AHG3, a member of PP2Cs, interacts with SnRK3.22 (PKS5)[33], which may participate in the regulation of PKS5 kinase activity. SOS3 and SCaBP8 bind to Ca²⁺ ions, and then recruit SOS2 to the plasma membrane and activate it. SOS2 activity results in the activation of the PM Na⁺/H⁺ antiporter SOS1, and reduced PKS5 activity triggers the activation of PM H⁺-ATPase by binding to 14-3-3ω, which provides SOS1 with the proton gradient needed to drive Na⁺ transportation (Fig. 7i). However, it is not clear whether and how calcium and calcium-binding protein SCaBPs/CBLs play roles in regulating PKS5 and SOS2 under normal growth conditions, and whether and how they function in deactivation of PKS5 under salt stress.

Previous studies suggest that J3 (DnaJ homolog 3) interacts with and represses PKS5 kinase activity under the normal growth condition, which in turn activates the PM H⁺-ATPase activity[29]. J3 interacts with the regulatory domain of PKS5 (PKS5C), and 14-3-3 proteins interacted with the kinase domain of PKS5 (PKS5N), which suggests that 14-3-3 proteins may directly repress PKS5 kinase activity, whereas J3 may change the structure of PKS5 and then represses the kinase activity of PKS5. However, it is not fully understood whether 14-3-3 and J3 repress PKS5 in different times or stress conditions or they work together to repress PKS5.

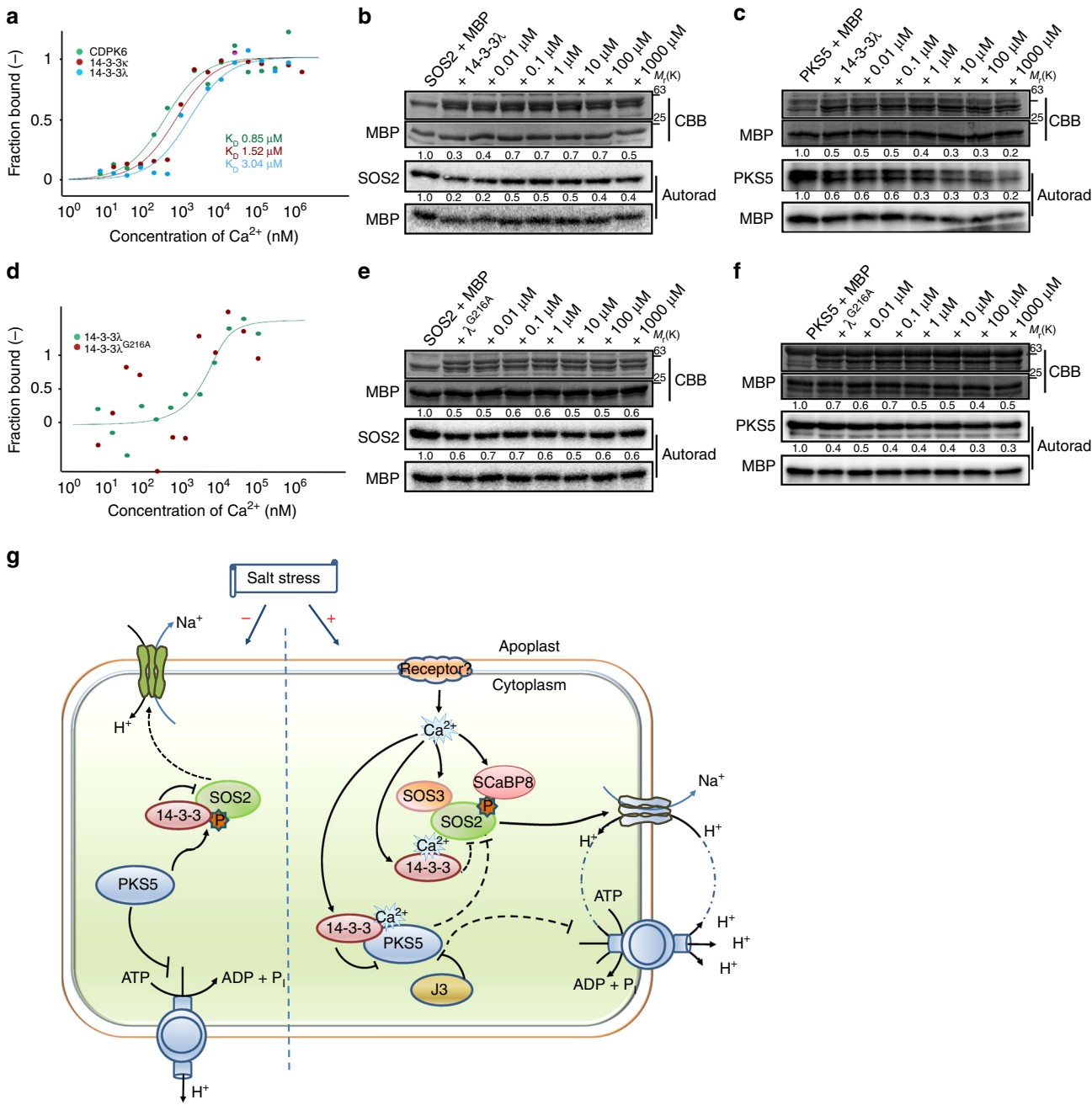

**Fig. 7** 14-3-3-mediated regulation of SOS2/PKS5 kinase activity by calcium ions. **a** MST assay showing that 14-3-3λ and κ interact with calcium ions. CDPK6 served as a positive control. Data are representative of three independent biological experiments. **b** In vitro kinase assay showing that $Ca^{2+}$ decreases the 14-3-3-mediated repression of SOS2. **c** In vitro kinase assay showing that $Ca^{2+}$ enhances the 14-3-3-mediated repression of PKS5. **d** MST assay of the $Ca^{2+}$-binding affinity of 14-3-3λ$^{G216A}$. Data are representative of three independent biological experiments. **e** In vitro kinase assay showing that the 14-3-3λ$^{G216A}$ mutant does not significantly repress SOS2 activity in the presence of $Ca^{2+}$. **f** In vitro kinase assay showing that the 14-3-3λ$^{G216A}$ mutant does not significantly repress PKS5 activity in the presence of $Ca^{2+}$. **g** Model illustrating how 14-3-3 proteins decode a calcium signal to enhance the salt tolerance of plants. Under normal growth conditions, PKS5 phosphorylates SOS2$^{Ser294}$ and represses SOS2 activity. Upon salt stress, 14-3-3 proteins decode a salt-induced calcium signal, repress PKS5, and release SOS2 to activate PM $H^+$-ATPase and SOS1, respectively. For **b** and **c**, about 1.0 μg of 14-3-3λ protein was incubated with or without $CaCl_2$ at a final concentration of 500 μM and 5 mM EGTA at room temperature for 30 min before the kinase assay. MBP was used as a substrate. CBB Coomassie brilliant blue, Autorad autoradiograph. Source data of **a**–**f** are provided in the Source Data file

To date, many studies have shown that 14-3-3 proteins play crucial roles in a variety of biological processes, in plants, and are key regulators of various signal transduction pathways. For instance, they transduce the cold signal into the nucleus and thereby induce the expression of cold responsive genes in Arabidopsis[34], and transduce the BR signal by localizing BZR1 to the nucleus in Arabidopsis and rice (*Oryza sativa*)[35]. In this study, we found that 14-3-3 proteins transmit salt-induced calcium signals to SOS2/PKS5 kinases and coordinately relieve/inhibit their activity, which triggers Na$^+$/H$^+$ antiporter and PM H$^+$-ATPases activity, and thereby maintains sodium ion homeostasis under salt stress.

In Arabidopsis, 8 out of 13 14-3-3 homologs, including 14-3-3λ, κ, and ω, were reported to contain an EF-hand motif[31], supporting the notion that calcium signals regulate 14-3-3-mediated SOS2 and PKS5 protein kinase inhibition. Activation of TMPK1 kinase by calcium ions requires 14-3-3ω[36] and 14-3-3ω binds calcium at its highly conserved C-terminus[32]. A mutation in the 14-3-3λ EF-hand motif that abolished its calcium-binding activity did not rescue the 14-3-3 mutant phenotype and removed the calcium-dependent and 14-3-3λ-mediated SOS2 and PKS5 kinase repression, demonstrating that calcium binding is essential for 14-3-3λ-mediated regulation of SOS2 and PKS5 activity. 14-3-3ω activates PM $H^+$-ATPases by binding to the penultimate C-terminal phosphorylated $Thr^{947}$, and a calcium signal may also be required for this regulation[37,38]. Phosphorylation of AHA2 $Ser^{931}$ by PKS5 blocks this binding, thereby inhibiting AHA2 activity[24]. However, 14-3-3ω does not interact with PKS5. These results indicate that the salt-induced calcium signal also coordinately regulates different 14-3-3 proteins to activate PM $H^+$-ATPase.

14-3-3λ and κ bind to the junction domain of SOS2 and phosphorylation of $Ser^{294}$ enhances this binding[21]; however, 14-3-3 proteins interact with the kinase domain of PKS5, which contains no conserved 14-3-3-binding motif, suggesting that 14-3-3 proteins play different roles in regulating SOS2 and PKS5 activity, and that the underlying regulatory mechanism may differ.

Interestingly, the 14-3-3λκ double mutant displayed high levels of $Na^+$ efflux (Supplementary Fig. 6f, g) and low levels of $H^+$ efflux (Supplementary Fig. 4e, f). Consistent with this observation, the mutant is more tolerant to sodium[21] and more sensitive to alkaline conditions than the wild type. However, activation of PM $H^+$-ATPase is required for enhancing $Na^+/H^+$ antiporter activity. We compared the salt sensitivity of 14-3-3λ-overexpression lines in pks5-4 with the 14-3-3λ-overexpression lines in the wild type (BigM), and found that the overexpression of 14-3-3λ in BigM resulted in the transgenic plants were more sensitive to salt treatment compared with that of BigM (Supplementary Fig. 7a, b). However, overexpressing 14-3-3λ in pks5-4 could not rescue the salt-sensitive phenotype of pks5-4 (Supplementary Fig. 7a, b), although the 14-3-3λ protein levels were similar in these overexpression lines (Supplementary Fig. 7c).These results suggest that a certain degree of PM $H^+$-ATPase activity is enough to provide the driving force for PM $Na^+/H^+$ antiporter activity and that SOS2 is more important for this activation.

Taken together, these findings suggest that switch-like 14-3-3 proteins bind to and repress SOS2 or PKS5 in the absence or presence of salt and that these protein–protein interactions are associated with an increased 14-3-3 calcium-binding affinity, which is critical for the plant's ability to adapt to salt stress.

## Methods

**Plant materials and growth conditions**. Arabidopsis Col-0 and Col erecta105 (BigM) plants were used in this study, pks5-3 and pks5-4 are in BigM background, and the others are in Col-0 background. The reported materials pks5-1, pks5-3, pks5-4, 14-3-3λκ, Pro35S:6×Myc-SOS2 in Col-0, Pro35S:6×Myc-SCaBP8 in Col-0, and Pro35S:Flag-HA-14-3-3λ in Col-0 were described previously[14,21,24,29]. The Pro35S:6×Myc-SOS2/SCaBP8 in BigM/pks5-1/pks5-3/pks5-4 and Pro35S:6×Myc-SOS2$^{S294A}$/Pro35S:Flag-HA-14-3-3λ in pks5-4 plants were generated by crossing, and two homozygous transgenic lines were detected by anti-C-Myc (CWBIO 01217/10153, 1/3000), anti-HA (Sigma-Aldrich H3663, 1/3000), or anti-Actin (CWBIO 01265/60205, 1/5000), and used for further studies. All the plants were grown at 22–24 °C under a 16-light and 8-h dark photoperiod. Seeds were imbibed at 4 °C for 2 days and geminated on Murashige–Skoog (MS) Petri dishes containing 0.5% phytagel and 2.0% sucrose in a light growth chamber.

**Mass spectrometry analysis**. The 10-day-old Pro35S:Flag-HA-SOS2 seedlings (about 5 g) in the sos2-2-mutant background[21] grown on MS Petri dishes were collocated and ground to a fine powder in liquid nitrogen. The plant material was incubated in IP buffer for 30 min on ice (10 mM Tris, pH 7.5, 0.5% Nonidet P-40, 2 mM EDTA, 150 mM NaCl, 1 mM PMSF, and 1% protease inhibitor cocktail, Roche) and centrifuged at 12,000 × g at 4 °C for 10 min to remove cell debris. The

anti-HA-conjugated agarose (Sigma-Aldrich) was used for HA-SOS2 immuno-precipitation. The SOS2 protein and its interacting proteins were then analyzed by a LTQ ORBITRAP Velos mass spectrometer (ThermoFisher Scientific). Database searches were performed on an in-house Mascot server (Matrix Science Ltd.) against International Protein Index (IPI) Arabidopsis database. The matched peptides were filtered with 0.05 significance threshold. The Mascot Percolator scores were used for all peptides.

To confirm the phosphorylation site of SOS2$^{Ser294}$ site by PKS5, samples containing 15 μg of His-tagged SOS2 and His-tagged PKS5 retained on beads. Total proteins were incubated in 30 μL of kinase buffer (20 mM Tris–HCl, pH 8.0, 1 mM DTT, 5 mM $MgCl_2$, and 10 μM ATP) at 30 °C for 30 min, His-tagged PKS5 were removed by a 5-min centrifugation at 500 × g at 4 °C. His-tagged SOS2 were collected and analyzed by LC–MS/MS.

**Plasmids construction and recombinant proteins**. To generate recombinant protein expression plasmid constructs, the coding sequences of SOS2-JD$^{S294A}$, 14-3-3λ, 14-3-3κ, and 14-3-3ω were amplified by PCR from Arabidopsis cDNA, and the resulting cDNA fragments were cloned into the BamHI and SalI sites of pGEX-6p-1 vector. The GST fusion recombinant proteins were expressed in E. coli DE3 strains. The bacteria were grown in liquid LB medium at 37 °C to an $OD_{600}$ of 1.0–1.5. The recombinant proteins were induced by 0.2 mM isopropyl β-D-1-thiogalactopyranoside (IPTG) and purified with glutathione sepharose (MCLAB). His-tagged SOS2 and PKS5 were purified by nickel–nitrilotriacetic acid agarose (MCLAB).

**Yeast two-hybrid assays**. To analyze the interaction between PKS5 and SOS2 or 14-3-3λ, the coding sequences of PKS5, PKS5N, PKS5JD, and PKS5C were cloned into the pGBKT7 vector, and the resulting vectors were co-transformed with pGADT7-SOS2 or pGADT7-14-3-3λ into the yeast strain AH109. To determine the interaction between 14-3-3s and AHA2 C100$^{T947D}$, the coding sequences of 14-3-3κ and 14-3-3ω were cloned into pGAKT7 vector, the AHA2 C100$^{T947D}$ coding sequence was cloned into pGBKT7 vector, and the resulting vectors were transformed into AH109. SC/-Trp/-Leu or SC/-Trp/-Leu/-His medium were used to detect the interactions.

**In vitro kinase assays**. An in vitro kinase assay was performed as described below. The recombinant proteins were purified from E. coli DE3 strains and the Myc-tagged proteins were immunoprecipitated from transgenic plants. For the kinase reaction, 25 μL of reaction mixtures with 1 μCi of [γ-$^{32}$P] ATP in kinase buffer (20 mM Tris–HCl, pH 8.0, 1 mM DTT, 5 mM $MgCl_2$, and 10 μM ATP) were prepared. The mixtures were kept at 30 °C for 30 min and 6 × SDS loading buffer was used to stop the reactions. Proteins were separated by a 10% or 12% SDS–PAGE gel, the phosphorylation signals were visualized by a Typhoon 9410 imager and quantified by ImageJ software.

**Bimolecular fluorescence complementation**. BiFC assays were performed using N. benthamiana as described below. The coding sequences of PKS5 and SOS2 were cloned into pSPYNE(R)173 and pSPYCE(M) vectors, respectively. The resulting constructs were transformed into Agrobacterium strain GV3101 and then infiltrated into N. benthamiana leaves. Three days later, the YFP fluorescence was detected by Leica confocal.

**LUC assay**. The coding sequences of PKS5 and AHA2 C100$^{T947D}$ were cloned into the pCAMBIA-nLuc vectors, the coding sequences of 14-3-3κ and 14-3-3ω were cloned into the pCAMBIA-cLuc vectors, and the nLuc-SOS2 and cLuc-14-3-3λ has been described[21]. All these constructs were introduced into Agrobacterium strain GV3101, and then infiltrated into N. benthamiana leaves as described below. Two days after infiltration, leaves incubated with 1 mM luciferin, and luminescence was recorded with a low-light cooled charge-coupled device camera. For NaCl treatment, leaf discs incubated with 1 mM luciferin containing 200 mM NaCl in a 96-well plate for 5–30 min, and the Luc signal was collected by the GLOMAX 96 microplate luminometer.

**SOS recruitment system**. SRS assays were performed using S. cerevisiae strain JP837, which is a derivative of AXT3K (Dena1:HIS3:ena4, nha1:LEU2, and nhx1:KanMX) as described below. Briefly, the coding sequences of SOS2$^{T168D/Δ308/S294A}$ and SOS2$^{T168D/Δ308/S294D}$ were cloned into p416GPD vectors, PKS5, PKS5-3 and PKS5-4 were cloned into p414GPD vectors, 14-3-3λ and ω were cloned into p415GPD vectors. The resulting plasmids were transformed into S. cerevisiae strain JP837 by a standard lithium-polyethylene glycol method and at least two positive clones were used for further studies. Salt tolerance of yeast were detected on AP medium (1 mM KCl, 8 mM phosphoric acid, 0.2 mM $CaCl_2$, 2 mM $MgSO_4$, 10 mM L-Arg, 2% glucose, plus trace elements, adjust to pH 6.5 with Arg) with or without different concentration of NaCl, and grown for 3–4 days at 28 °C.

**In vitro pull-down and in vivo Co-immunoprecipitation assays**. For in vitro pull-down assays, all recombinant proteins were purified from E. coli DE3 strains. Briefly, GST-tagged SOS2-JD protein retained on the beads incubated

with His-tagged PKS5 in kinase buffer with ATP at 30 °C for 30 min. After removed His-tagged PKS5 by washing three times with PBS buffer, GST-tagged SOS2-JD protein were incubated with His-tagged 14-3-3λ in binding buffer (2 mM DTT, 10 mM MgCl₂ and 20 mM Tris–HCl, pH 7.2) at 4 °C for 3 h, and then the beads were wash for three times with PBS buffer. The pulled-down proteins were detected by immunoblot analysis using an anti-14-3-3 antibody (Santa Cruz SC-33752, 1/2000).

For in vivo Co-immunoprecipitation assays, stable transgenic plants were used to detect the interaction between 14-3-3 proteins and SOS2/PKS5 and the phosphorylation of SCaBP8$^{Ser237}$ in planta. Total proteins were extracted from Arabidopsis using IP buffer (10 mM Tris, pH 7.5, 2 mM EDTA, 150 mM NaCl, 0.5% Nonidet P-40, 1 mM phenylmethylsulfonyl fluoride, and 1% protease inhibitor cocktail; Roche). After removed cellular debris, the supernatant was incubated with anti-C-Myc agarose (Sigma-Aldrich) at 4 °C for 3 h, and then the beads were washed for five times with IP buffer. The associated proteins were analyzed by immunoblots and detected with anti-C-Myc (CWBIO 01217/10153, 1/3000) and anti-P-SC8 (made by AbMart, 1/2000) antibodies.

**RT-PCR analysis**. Total RNA was extracted from the transgenic *N. benthamiana* plants with Trizol reagent (Invitrogen). RNase-free DNase I (TaKaRa) was used to remove genomic DNA, and the RNA was reverse transcribed using M-MLV reverse transcriptase (Promega). Resulting cDNAs were used to detect the genes expression by RT-PCR analysis. The specific primers were listed in Supplementary Table 1.

**Salt and alkali sensitive assays**. For salt tolerance assays, 5-day-old seedlings grown on MS medium were transformed to medium containing 75, 100, or 125 mM NaCl. For alkali tolerance assays, 7-day-old seedlings were transformed to medium at pH 6.5 or 6.7, which was adjusted by 500 mM NaHCO₃. All the seedlings were grown in a light growth chamber at 22–24 °C under a 16-h light and 8-h dark photoperiod.

**Plasma membrane H⁺-ATPase activity assays**. The plasma membrane H⁺-ATPase activity assays were performed as described above. Plasma membrane-enriched vesicles were isolated from 5-week-old plants treated with 250 mM NaCl for 3 days. Fifty micrograms of plasma membrane protein was prepared for further measurement. When used, 500 ng of recombinant protein was incubated with plasma membrane protein at room temperature for 10 min. The activity of the H⁺-ATPase was reflected by the change in fluorescence of quinacrine (a pH-sensitive fluorescent probe), which was obtained by a Hitachi F-7500 imager.

**Measurement of Net Na+ and H+ fluxes with the NMT**. The measurement of Na⁺ and H⁺ fluxes was performed as described below. Net flux of Na⁺ and H⁺ were measured by NMT (YoungerUSA, LLC, Amherst, MA, USA) and imFluxes software. For Na⁺ flux, 7-day-old seedlings were treated in medium with 100 mM NaCl for 24 h. Pre-pulled and silanized microsensor (ø4.5 ± 0.5 μm, XY- CGQ -01) were filled with a backfilling solution (Na⁺: 250 mM NaCl; H⁺: 15 mM NaCl+40 mM KH₂PO₄, pH 7.0) to a length of nearly 1.0 cm and then 50 μm columns of selective liquid ion-exchange cocktails (LIXs, Na⁺: XY-SJ-Na; H⁺: XY-SJ-H. Younger, USA) were filled from the tip. Prior to the flux measurement, the seedling was incubated in the measuring solution (Na⁺: 0.1 mM CaCl₂, 0.1 mM KCl, 0.5 mM NaCl, and 0.3 mM MES, adjusted pH to 6.0 with Tris–HCl (pH 8.8); H⁺: 0.1 mM CaCl₂, 0.1 mM KCl, and 0.3 mM MES, adjusted pH to 7.0 with 1 M KOH) for 10 min. The microsensor was calibrated with 0.5 or 5 mM NaCl in calibration liquid (0.1 mM CaCl₂, 0.1 mM KCl, 0.3 mM MES, pH 6.0) for measuring Na⁺ flux; and with pH 5.5 or 6.5 in calibration liquid (0.1 mM CaCl₂, 0.1 mM KCl, 0.3 mM MES) for measuring H⁺ flux. Ion fluxes were measured in the zone about 300 μm from the tip. All the measurement results were exported by using the JCal V3.3 (a free MS Excel spreadsheet, youngerusa.com or xbi.org) and the consumables were provided by Xuyue (Beijing).

**MST assays**. The binding affinities of 14-3-3λ/κ with calcium ions were detected by Monolith NT.115 (Nanotemper Technologies). All recombinant proteins were purified from *E. coli* DE3 strains, GST-tagged 14-3-3λ/κ and GST-tagged CDPK6 were labeled in accordance with the manufacturer's procedure. A volume of 100 μL (10 μM) proteins were exchanged into a labeling buffer and labeled by dye NT-647-NHS at room temperature for 30 min in the dark. The labeled proteins were incubated with different concentration of calcium ions or ligandins for 10 min, and then loaded into silica capillaries (Polymicro Technologies). The samples were measured by Monolish NT.115 (NanoTemper Technologies) at 25 °C, 20% MST power and 20% LED power. The results were analyzed by OriginPro 9.0 software and MO. Affinity Analysis software (V2.2.4).

**Reporting summary**. Further information on experimental design is available in the Nature Research Reporting Summary linked to this article.

## Data availability
All supporting data from this study are available from the article and Supplementary Information files, or from the corresponding author upon request. The source data underlying Figs. 1c, d, 2a, b, e–h, 3b–d, f–h, 4a, c, e, 5a–c, 6b, c, e, f and 7a–f, and Supplementary Figs. 1c, d, 3b, d, 4b, c, e, f, 5a–h, and 6e–g are provided in the Source Data file.

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

## Acknowledgements

This work was supported by the National Basic Research Program of China (2015CB910202) and the National Natural Science Foundation of China (31430012, 31670260, 31861133005, U1706201).

## Author contributions

Y.G., Y.Y. and Z.Y. conceived and designed the research plans; Z.Y., C.W., Y.X. and X.L. performed the experiments; Y.G., Z.Y., C.S. and Y.Y. analyzed data. S.C. generated and analyzed MS/MS data; Y.G., Z.Y., C.S. and Y.Y. wrote the paper.

## Additional information

**Competing interests:** The authors declare no competing interests.

