## [Peer Review File · Nature Communications]

Reviewers' comments:

Reviewer #1 (Remarks to the Author):

Review on the manuscript by Yang et al entitled 'Calcium-activated 14-3-3 proteins as a molecular switch in salt stress tolerance'

General comments:

in their current ms the authors follow up their previous findings on the role of 14-3-3 proteins in SOS pathway during salt stress adaptation. Here they show that the key kinase SOS2 (CIPK24) is under control of the kinase PKS5 (CIPK11). The authors provide strong evidence that PKS5 negatively regulates SOS2 kinase activity through phosphorylation of SOS2 which enhances the interaction between SOS2 and inhibitory 14-3-3 proteins. By this PKS5 would negatively affect salt stress tolerance. Salt stress however reduces PKS5 kinase activity, which is mediated again by 14-3-3 interaction. PKS5 is known to inhibit pm ATPase activity and here he author provide additional evidence for a role of the kinase in alkaline stress tolerance. Finally the show that 14-3-3 lambda and kappa bind Ca²⁺ and thus act as additional Ca²⁺ sensors/decoders in the SOS pathway. The ms is well written and technically sound. The broad spectrum of techniques employed is impressive and experiments have been conducted with great accuracy as known from this lab.

Some findings/statements however require further explanation and raise the following questions/comments:

In their final model the authors conclude that 'Under normal growth conditions, PKS5 phosphorylates SOS2Ser294, which promotes the 14-3-3/SOS2 interaction that in turn inhibits the SOS2 kinase activity to limit SOS1 Na⁺/H⁺ antiporter activity to basal levels;'

Q/C: What is the role of Ca²⁺ in this context given that both CIPK24 and CIPK11 are activated by Ca²⁺ binding CBL proteins. Their role is not discussed here, but would be crucial for a mechanistic understanding of this process. Likewise the Ca²⁺ affinity determined for the 14-3-3 proteins would far exceed 'resting' Ca²⁺ levels as reported for plant cells.

The authors further state: 'Furthermore, PKS5 phosphorylates PM H⁺-ATPases, thus limiting their activity, which is required to provide the driving force for the Na⁺/H⁺ antiporter activity of SOS1.'

Q/C: Under 'non-stressed' growth conditions the proton ATPase is crucial for membrane potential maintenance as well as solute/nutrient uptake. Is PKS5 addressing distinct pm ATPases? and how does the J3 chaperone fits into the model?

Q/C: In the second part of the model – under salt stress – the role of CIPK11 interacting CBLs should be discussed, since upon Ca²⁺ binding they would counteract 14-3-3 mediated PKS5 inhibition.

Further questions/comments

1. Identification of SOS2-interacting proteins

Q/C: The MS data of interacting PKS5 proteins should be shown as suppl. information

The authors employed His-tagged SOS2K40N, a kinase-dead type mutant, and showed that SOS2K40N was phosphorylated by PKS5 (Fig. 1c).

Q/C: Why did the authors use a kinase dead mutant – what about phosphorylation of PKS5 by SOS2?

Q/C: Is there an effect of CBLs or Ca²⁺ on phosphorylation?

2. PKS5 enhances the interaction between SOS2 and 14-3-3 proteins

'SOS2-JD interacts with 14-3-3 λ , and that this interaction was enhanced by the addition of PKS5 particularly in the presence of kinase buffer containing ATP (Fig. 2a).'

Q/C: the authors should include SOS2-JD-S294A in this pull down experiment!!

'Higher 14-3-3 protein signal resulted from SOS2 immunoprecipitation in the pks5-3 and pks5-4 backgrounds than in the BigM control; however, less immunoprecipitated 14-3-3 protein signal was observed in the pks5-1 background than in Col-0 (Fig. 2b).'

Q/C: this finding is hard to follow from the figure; I suggest to quantify data by densitometric analysis.

'Split-luciferase complementation (LUC) imaging assays in *N. benthamiana* also indicated that the presence of PKS5 enhanced the interaction between SOS2 and 14-3-3 λ , and that the enhancement was greater when the constitutively active kinases PKS5-3 and PKS5-4 were used (Fig. 2c, d).

Q/C: The authors should include SOS2-S294A in the split LUC assay, as well as PKS5 kinase dead mutant as a control;

Q/C: What is the role of Ca²⁺ and CBLs in this type of interaction?

'NaCl-induced activation of SOS2 was repressed in pks5-3 and pks5-4 (Fig. 2e).'

Q/C: while the NaCl induced activation of SOS2 is convincingly shown, the reduced activity in pks5-3 and pks5-4 needs densitometric analysis to comprehend the authors' statement.

4. PKS5 negatively regulates salt tolerance in Arabidopsis'

'Salt-stressed roots exhibited a pronounced net Na⁺ efflux, however, efflux was significantly lower in pks5-3 and pks5-4 than in BigM (Supplementary Fig. 3c and d).'

Q/C: probably the authors refer to figs 3c and d ?!

'We next measured Na⁺ flux in the crossed lines, and found that SOS2S294A expression rescued the faulty Na⁺ flux of pks5-4, whereas expression of SOS2 did not (Fig. 3g, h).'

Q/C: Fig. 3g shows the opposite of 3h!!

5. Salt Stress Enhances the Interaction between 14-3-3 Proteins and PKS5

'Co-immunoprecipitation assays confirmed that PKS5 interacted with 14-3-3 λ , and that this interaction was induced by NaCl treatment (Fig. 4c).'

Q/C: The NaCl induction is hard to see. The authors should quantify the interaction densitometrically.

6. 14-3-3 Proteins Repress the Kinase Activity of PKS5

' We found that both 14-3-3 λ and 14-3-3 κ repressed the PKS5 transphosphorylation activity of SCaBP1 and also repressed PKS5 autophosphorylation.'

Q/C: Again, the authors should quantify the interaction densitometrically in a, b, c and test for significance.

' On pH 6.5 and 6.7 MS medium, root elongation was inhibited to a greater extent in the 14-3-3 $\lambda\kappa$ double mutant than in Col-0 (Supplementary Fig. S4a and b).'

Q/C: This effect should be most pronounced under salt stress and not under 'standard' growth conditions when ATPase activity is required for solute/nutrient transport.

` NMT measurement also indicated that the H⁺ efflux of 14-3-3 λκ double mutants was lower than that of Col-0 (Supplementary Fig. S4e and f).`

Q/C: If interpreting the data correctly there is no H⁺ efflux; rather increased H⁺ influx in the 14-3-3 mutant.

7. 14-3-3 Proteins Are Required for PKS5 Function in Alkaline Stress Tolerance

` These results further indicate that 14-3-3λ and κ regulate the alkaline stress response in Arabidopsis at least partly by repressing of PKS5 kinase activity.`

Q/C: How this finding relates to salt stress? In other words: is salt stress affecting pH homeostasis?

8. Decoding the Salt-induced Calcium Signal by 14-3-3 and SOS3/SCaBP8 Calcium Sensors

`...and found that, in contrast to the negative control (GST-tagged protein), GST-14-3-3λ and GST-14-3-3κ bound Ca²⁺ directly (Fig. 7a).`

Q/C: How was the free Ca²⁺ concentration determined/adjusted? The authors should include a positive control; e.g. EF-hand containing protein like CPK or CaM

Q/C: The number of replicates in Fig.7b,c is just n=2

Reviewer #2 (Remarks to the Author):

In this manuscript, the authors show that 14-3-3 proteins can inhibit SOS2 that is phosphorylated by PKS5. They also show that the 14-3-3 inhibits PKS5 and that those inhibitions depend on calcium concentration. Thus, the role of 14-3-3 proteins in the regulation of plasma membrane Na/H transporter under the salt stress condition in Arabidopsis is concluded. I think that basically these are interesting findings appealing the broad audience. However, I have some concerns over the results.

1). In the authors' model, 14-3-3 has dual roles in the opposite way in the salt signaling pathway: inhibition of repressor (PKS5) and inhibition of activator (SOS2). The 2nd role depends on the phosphorylation by the repressor in the 1st role.

Overexpression of 14-3-3λ induces salt sensitive phenotype in the previous report (ref#21), suggesting that the inhibition of SOS2 is valid even though PKS5 is suppressed by the overexpression of 14-3-3. This is a bit difficult to figure out. Comparison of salt sensitivity of 14-3-3-overexpression lines in pks5-4 (Fig.6) with that of 14-3-3-overexpression lines in Col-0 (ref#21) might provide more hints.

2). Pull-down experiment (Fig.2a).

PKS5 is also fused to His-tag. If His-PKS5 is degraded, the degradation bands may be overlapped on the 14-3-3 bands. The pulldown fraction with PKS5 but without 14-3-3 should be shown as a control.

3). I think that the autorad panel of MBP and SOS2 are opposite in Fig. 7d. If so, the effects of Ca on phosphorylation of MBP are relatively small.

4). I think that +GFP and +PKS5 are mislabeled in SuppleFig S2.

5). "Supplemental Fig.3c and d" should be "Fig.3c and d" (P8, L208)

Response to Reviewers' Comments

We would like to thank the reviewers again for their helpful evaluation of and comments and suggestions for our manuscript.

All comments have been addressed experimentally and additional modifications and explanations have been provided to strengthen our conclusions. These changes have allowed us to substantially improve the clarity of our manuscript for NC readers.

Below, we respond to all Reviewers' comments and questions. The reviewer comments are laid out below in thickened and italicized font and specific concerns have been numbered. Our response is given in normal font.

Response to reviewer 1:

In their final model the authors conclude that 'Under normal growth conditions, PKS5 phosphorylates SOS2^{Ser294}, which promotes the 14-3-3/SOS2 interaction that in turn inhibits the SOS2 kinase activity to limit SOS1 Na⁺/H⁺ antiporter activity to basal levels;'

Q/C: What is the role of Ca²⁺ in this context given that both CIPK24 and CIPK11 are activated by Ca²⁺ binding CBL proteins. Their role is not discussed here, but would be crucial for a mechanistic understanding of this process. Likewise the Ca²⁺ affinity determined for the 14-3-3 proteins would far exceed 'resting' Ca²⁺ levels as reported for plant cells.

Response:

We do not know whether the Ca²⁺ plays a role through the Ca²⁺ binding CBL proteins in activation or deactivation of CIPK11/PKS5 or CIPK24/SOS2 under the normal growth condition. Therefore we did not discuss the SCaBP/CBL proteins in the manuscript.

In yeast, we previously reported that PKS5 and SCaBP1 can inhibit AHA2 activity, but none of them alone represses AHA2 (Fuglsang et al., 2007). However, SCaBP1/CBL2 mutants do not show any phenotypic change under salt treatment. Moreover, it has been shown that CBL2 and CBL3 proteins locate at the tonoplast membrane (Tang et al., 2015). We found that the addition of Ca²⁺ and SCaBP1 did not affect the phosphorylation of SOS2 by PKS5 in an *in vitro* kinase assay (Supplementary Fig. 1d), and the expression of SCaBP1 had no significant effect on the PKS5-mediated SOS2 and 14-3-3 λ interaction in LUC imaging assay (Supplementary Fig. 2g). In the revised manuscript, we discuss that other

SCaBP/CBL may be involved in activation/deactivation of PKS5 coordinately with 14-3-3 proteins.

Previous studies reported that the concentration of cytosolic free Ca^{2+} can reach 1000 nM when treated with NaCl in Arabidopsis (Tracy et al., 2008). Changes in cytosolic free Ca^{2+} in response to salt treatments were detected *in vivo* in intact whole Arabidopsis seedlings, and transient elevations of cytosolic free Ca^{2+} to about 1.5 μM were observed (Knight et al., 1997). In this study, we used different Ca^{2+} concentrations to suggest that the higher Ca^{2+} concentration induces the interaction between PKS5 and 14-3-3 and represses the interaction between SOS2 and 14-3-3 under salt stress.

The authors further state: ‘Furthermore, PKS5 phosphorylates PM H^+ -ATPases, thus limiting their activity, which is required to provide the driving force for the Na^+/H^+ antiporter activity of SOS1.’

Q/C: Under ‘non-stressed’ growth conditions the proton ATPase is crucial for membrane potential maintenance as well as solute/nutrient uptake. Is PKS5 addressing distinct pm ATPases? and how does the J3 chaperone fits into the model?

Response:

Salt stress induces the increase in PM H^+ -ATPase activity, which provides the driving force for the Na^+/H^+ antiporter activity. It is important that under ‘non-stressed’ growth conditions the proton ATPase is crucial for membrane potential maintenance as well as solute/nutrient uptake. Therefore, the precise regulation of PM H^+ -ATPase activity is critical for plant growth and stress response. We do not know whether PKS5 participates in the regulation of other pm ATPases except H^+ -ATPase at present.

As suggested, we discussed J3 in detail in the revised manuscript. J3 represses PKS5 activity under salt-alkaline condition. However, we do not know whether J3 and 14-3-3 repress PKS5 in different times, or stress conditions or they work together to repress PKS5.

Q/C: In the second part of the model – under salt stress – the role of CIPK11 interacting CBLs should be discussed, since upon Ca^{2+} binding they would counteract 14-3-3 mediated PKS5 inhibition.

Response:

We do not know whether the Ca^{2+} plays a role through the Ca^{2+} binding CBL proteins in deactivation of CIPK11 under salt stress. Therefore we did not discuss the SCaBP/CBL proteins in the manuscript.

In the revised manuscript, we discuss that other SCaBP/CBL may be involved in activation/deactivation of PKS5 coordinately with 14-3-3 proteins.

In an independent study, we identified that one of SCaBPs/CBLs directly interacted with and repressed PM H⁺-ATPase activity, and enhanced the interaction between PKS5 and AHA2 under normal condition (non-stress); under salt-alkaline stress, the stress-induced increase in cytosolic Ca²⁺ negatively affected the SCaBP-mediated interaction between PKS5 and AHA2. This manuscript is revised at the Plant Cell. I wish that the results could provide some ideas how SCaBP/CBL and 14-3-3 coordinately regulate PKS5 activity.

Further questions/comments

1. Identification of SOS2-interacting proteins

Q/C: The MS data of interacting PKS5 proteins should be shown as suppl.

Information

Response:

As suggested, we added the MS data in Supplementary Fig. 1a.

Q/C: Why did the authors use a kinase dead mutant – what about phosphorylation of PKS5 by SOS2?

Response:

In order to avoid an interference caused by auto-phosphorylation of SOS2, a kinase-dead type SOS2 mutant (His-tagged SOS2^{K40N}, Liu et al., 2000) is used for the kinase assay. As suggested, we detected the phosphorylation of PKS5 by SOS2, and found that SOS2 could not phosphorylate the kinase-dead type mutant PKS5^{K50N} (Supplementary Fig. 1c).

Q/C: Is there an effect of CBLs or Ca²⁺ on phosphorylation?

Response:

As suggested, we performed an *in vitro* kinase assay to detect the effect of SCaBP1 and Ca²⁺ on the phosphorylation of SOS2 by PKS5. SOS2-JD was used as the substrate of PKS5. The results showed that SCaBP1 and 10 μM Ca²⁺ had no significant effect on the phosphorylation of SOS2 by PKS5 (Supplementary Fig. 1d).

2. PKS5 enhances the interaction between SOS2 and 14-3-3 proteins

Q/C: the authors should include SOS2-JD-S294A in this pull down experiment!!

Response:

As suggested, we added the SOS2-JD-S294A in the pull down experiment, as shown in the revised Fig. 2a.

Q/C: *'Higher 14-3-3 protein signal resulted from SOS2 immunoprecipitation in the pks5-3 and pks5-4 backgrounds than in the BigM control; however, less immunoprecipitated 14-3-3 protein signal was observed in the pks5-1 background than in Col-0 (Fig. 2b).'* *This finding is hard to follow from the figure; I suggest to quantify data by densitometric analysis.*

Response:

To make such conclusion, we have repeated the similar experiments for quite few times. In order to better explain the experimental results, we provide two other biological replicate experiments in the revised manuscript, and they showed similar results. As suggested, we quantify data by densitometric analysis (Fig. 2b).

Q/C: *The authors should include SOS2-S294A in the split LUC assay, as well as PKS5 kinase dead mutant as a control;*

Response:

As suggested, we added the SOS2-JD-S294A and PKS5 kinase dead mutant (PKS5^{K50N}) in the split LUC assay, as shown in the revised Supplementary Fig. 2e.

Q/C: *What is the role of Ca²⁺ and CBLs in this type of interaction?*

Response:

As suggested, we detected the effect of SCaBP1 on the interaction of SOS2 and 14-3-3λ in the split LUC assay, and found that the expression of SCaBP1 had no significant effect on this interaction (Supplementary Fig. 2g). MST assay indicated that a certain degree of Ca²⁺ could decrease the interaction between SOS2 and 14-3-3λ (Fig. 7b).

Q/C: *while the NaCl induced activation of SOS2 is convincingly shown, the reduced activity in pks5-3 and pks5-4 needs densitometric analysis to comprehend the authors' statement.*

Response:

To make such conclusion, we have repeated the similar experiments for quite few times. In order to better explain the experimental results, we provide two other

biological replicate experiments in the revised manuscript, and they showed similar results. As suggested, we quantify data by densitometric analysis (Fig. 2e, f).

4. *PKS5 negatively regulates salt tolerance in Arabidopsis*

‘Salt-stressed roots exhibited a pronounced net Na⁺ efflux, however, efflux was significantly lower in pks5-3 and pks5-4 than in BigM (Supplementary Fig. 3c and d).’

Q/C: probably the authors refer to figs 3c and d ?!

Response:

Thanks to point out this mistake. We checked and corrected ‘Supplementary Fig. 3c and d’ to ‘Fig. 3c and d’ in the revised manuscript.

‘We next measured Na⁺ flux in the crossed lines, and found that SOS2^{S294A} expression rescued the faulty Na⁺ flux of pks5-4, whereas expression of SOS2 did not (Fig. 3g, h).’

Q/C: Fig. 3g shows the opposite of 3h!!

Response:

Thanks to point out this mistake. We incorrectly marked S2 and S2^{SA} in opposite order in Fig. 3g. In the revised manuscript, we corrected this mistake.

5. *Salt Stress Enhances the Interaction between 14-3-3 Proteins and PKS5*

‘Co-immunoprecipitation assays confirmed that PKS5 interacted with 14-3-3λ, and that this interaction was induced by NaCl treatment (Fig. 4c).’

Q/C: The NaCl induction is hard to see. The authors should quantify the interaction densitometrically.

Response:

To make such conclusion, we have repeated the similar experiments for quite few times. In order to better explain the experimental results, we provide two other biological replicate experiments in the revised manuscript, and they showed similar results. As suggested, we quantify data by densitometric analysis (Fig. 4c).

6. *14-3-3 Proteins Repress the Kinase Activity of PKS5*

‘ We found that both 14-3-3λ and 14-3-3κ repressed the PKS5 transphosphorylation activity of SCaBP1 and also repressed PKS5 autophosphorylation.’

Q/C: Again, the authors should quantify the interaction densitometrically in a, b, c and test for significance.

Response:

To make such conclusion, we have repeated the similar experiments for quite few times. In order to better explain the experimental results, we provide two other biological replicate experiments in the revised manuscript, and they showed similar results. As suggested, we quantify data by densitometric analysis (Fig. 5).

‘ On pH 6.5 and 6.7 MS medium, root elongation was inhibited to a greater extent in the 14-3-3 $\lambda\kappa$ double mutant than in Col-0 (Supplementary Fig. S4a and b).’

Q/C: This effect should be most pronounced under salt stress and not under ‘standard’ growth conditions when ATPase activity is required for solute/nutrient transport.

Response:

Our previous studies found that the 14-3-3 $\lambda\kappa$ double mutant is more tolerant to salt treatment in soil than that of Col-0, and overexpression of 14-3-3 leads that the transgenic plants are more sensitive to salt treatment than that of wild type (Zhou et al., 2014). The salt phenotype of 14-3-3 double mutant is not as pronounced as that under high pH contention. We think the reason is that SOS2 plays a major role in regulating SOS1 activity under salt stress, although PM H⁺-ATPase activity is also required for this regulation. However, under high pH condition, the regulation of PM H⁺-ATPase activity is more important for the plant growth, because a higher PM H⁺-ATPase activity provides a better micro-environment (with lower pH) surrounding root (Fuglsang et al, 2007).

‘ NMT measurement also indicated that the H⁺ efflux of 14-3-3 $\lambda\kappa$ double mutants was lower than that of Col-0 (Supplementary Fig. S4e and f).’

Q/C: If interpreting the data correctly there is no H⁺ efflux; rather increased H⁺ influx in the 14-3-3 mutant.

Response:

Under normal (non-stress) conditions, the net H⁺ flux measured by non-invasive microsensing system (NMS) is shown as influx due to the combined effect of efflux and influx of H⁺. The similar test method was reported in Xue et al. (2018), Han et al. (2017), He et al. (2015), Hao et al. (2012), and Xu et al. (2012).

7. 14-3-3 Proteins Are Required for PKS5 Function in Alkaline Stress Tolerance

‘ These results further indicate that 14-3-3 λ and κ regulate the alkaline stress response in Arabidopsis at least partly by repressing of PKS5 kinase activity. ’

Q/C: *How this finding relates to salt stress? In other words: is salt stress affecting pH homeostasis?*

Response:

Yes, salt stress affects pH homeostasis.

Previous studies found that plants PM H⁺-ATPase activity increased significantly after NaCl treatment (Yang et al., 2010). When plants were treated with 75 mM NaCl, an increase of H⁺ efflux was seen in the root by the MIFE technique, and a decrease in apoplast pH in the root was seen as a fluorescence change by confocal microscopy (Fuglsang et al, 2007; Yang et al., 2010). These results suggest that salt stress affecting pH homeostasis through increasing the PM H⁺-ATPase activity. In this study, 14-3-3 λ and κ could repress PKS5-mediated regulation of PM H⁺-ATPase under salt stress, which leads to an increase of PM H⁺-ATPase and provides the driving force for Na⁺/H⁺ antiporter SOS1.

8. Decoding the Salt-induced Calcium Signal by 14-3-3 and SOS3/SCaBP8 Calcium Sensors

Q/C: *How was the free Ca²⁺ concentration determined/adjusted? The authors should include a positive control; e.g. EF-hand containing protein like CPK or CaM*

Response:

We used EGTA to make a free Ca²⁺ concentration and provided this information in the method. As suggested, we used CDPK6 as a positive control in MST assay, as shown in Fig. 7a.

Q/C: *The number of replicates in Fig. 7b,c is just n=2*

Response:

To make such conclusion, we have repeated the similar experiments for quite few times. In order to better explain the experimental results, we showed two other biological replicate experiments and the results were similar.

Response to reviewer 2:

1. In the authors' model, 14-3-3 has dual roles in the opposite way in the salt signaling pathway: inhibition of repressor (PKS5) and inhibition of activator (SOS2). The 2nd role depends on the phosphorylation by the repressor in the 1st role.

Overexpression of 14-3-3λ induces salt sensitive phenotype in the previous report (ref#21), suggesting that the inhibition of SOS2 is valid even though PKS5 is suppressed by the overexpression of 14-3-3. This is a bit difficult to figure out. Comparison of salt sensitivity of 14-3-3-overexpression lines in pks5-4 (Fig.6) with that of 14-3-3-overexpression lines in Col-0 (ref#21) might provide more hints.

Response:

We thanks for the suggestion. We compared the salt sensitivity of 14-3-3λ-overexpression lines in *pks5-4* with 14-3-3λ-overexpression lines in BigM, and found that overexpression of 14-3-3λ in BigM leading to a salt-sensitive phenotype compared with BigM in MS medium containing 100 mM NaCl, which is consistent with pervious results (Zhou et al., 2014; Supplementary Fig. 7a). However, overexpressing 14-3-3λ in *pks5-4* could not significantly rescue the salt-sensitive phenotype of *pks5-4* (Supplementary Fig. 7a).

We think the reason is that SOS2 plays a major role in regulating SOS1 activity under salt stress, although PM H⁺-ATPase activity is also required for this regulation. However, under high pH condition, the regulation of PM H⁺-ATPase activity by PKS5 is more important for the plant growth, because a higher PM H⁺-ATPase activity provides a better micro-environment (with lower pH) surrounding root (Fuglsang et al, 2007).

2.Pull-down experiment (Fig.2a).

PKS5 is also fused to His-tag. If His-PKS5 is degraded, the degradation bands may be overlapped on the 14-3-3 bands. The pulldown fraction with PKS5 but without 14-3-3 should be shown as a control.

Response:

Thanks for this suggestion. To avoid the degradation bands of His-PKS5, we used anti-14-3-3 antibody to detect the binding of 14-3-3λ with PKS5, as shown in Fig. 2a.

3.I think that the autorad panel of MBP and SOS2 are opposite in Fig. 7d. If so, the effects of Ca on phosphorylation of MBP are relatively small.

Response:

We apologized for this mistake and thanks for this reviewer to point out this mistake. The addition of Ca^{2+} could decrease the phosphorylation of MBP by SOS2, which had more significant effect on the autophosphorylation of SOS2. We modified the notion in the revised manuscript.

To make such conclusion, we have repeated the similar experiments for quite few times. In order to better explain the experimental results, we provide two other biological replicate experiments in the revised manuscript, and they showed similar results. As suggested, we quantify data by densitometric analysis.

4.I think that +GFP and +PKS5 are mislabeled in SuppleFig S2.

Response:

We apologized for this mistake and thanks for your correction. We corrected it in the revised manuscript.

5. “Supplemental Fig.3c and d” should be “Fig.3c and d” (P8, L208)

Response:

We apologized for this mistake and thanks for your correction. We corrected it in the revised manuscript.

References:

Han, X., Yang, Y., Wu, Y., Liu, X., Lei, X., and Guo, Y. (2017). A bioassay-guided fractionation system to identify endogenous small molecules that activate plasma membrane H^+ -ATPase activity in Arabidopsis. *J. Exp. Bot.* 68, 2951-2962.

Liu, J., Ishitani, M., Halfter, U., Kim, C.S., and Zhu, J.K. (2000). The Arabidopsis thaliana SOS2 gene encodes a protein kinase that is required for salt tolerance. *Proc. Natl. Acad. Sci. USA* 97, 3730-3734.

Xue, Y., Yang, Y., Yang, Z., Wang, X., and Guo, Y. (2018). VAMP711 Is Required for Abscisic Acid-Mediated Inhibition of Plasma Membrane H^+ -ATPase Activity. *Plant Physiol.* 178, 1332-1343.

Yang, Y., Qin, Y., Xie, C., Zhao, F., Zhao, J., Liu, D., Chen, S., Fuglsang, A.T., Palmgren, M.G., Schumaker, K.S., *et al.* (2010). The Arabidopsis chaperone J3 regulates the plasma membrane H^+ -ATPase through interaction with the PKS5 kinase. *Plant Cell* 22, 1313-1332. Zhou, H., Lin, H., Chen, S., Becker, K., Yang, Y., Zhao, J., Kudla, J., Schumaker, K.S., and

Guo, Y. (2014). Inhibition of the Arabidopsis salt overly sensitive pathway by 14-3-3 proteins. *Plant Cell* 26, 1166-1182.

Fuglsang, A.T., Guo, Y., Cuin, T.A., Qiu, Q., Song, C., Kristiansen, K.A., Bych, K., Schulz, A., Shabala, S., Schumaker, K.S., et al. (2007). Arabidopsis protein kinase PKS5 inhibits the plasma membrane H⁺-ATPase by preventing interaction with 14-3-3 protein. *Plant Cell* 19, 1617-1634.

Kudla, J., Xu, Q., Harter, K., Griessem, W., and Luan, S. (1999). Genes for calcineurin B-like proteins in Arabidopsis are differentially regulated by stress signals. *Proc. Natl. Acad. Sci. USA* 96, 4718-4723.

Knight, H., Trewavas, A.J., and Knight, M.R. (1997). Calcium signalling in *Arabidopsis thaliana* responding to drought and salinity. *Plant J.* 12, 1067-1078.

Tang, R.J., Zhao, F.G., Garcia, V.J., Kleist, T.J., Yang, L., Zhang, H.X., and Luan, S. (2015). Tonoplast CBL-CIPK calcium signaling network regulates magnesium homeostasis in Arabidopsis. *Proc. Natl. Acad. Sci. USA* 112, 3134-3139.

Tracy, F.E., Gilliam, M., Dodd, A.N., Webb, A.A., and Tester, M. (2008). NaCl-induced changes in cytosolic free Ca²⁺ in *Arabidopsis thaliana* are heterogeneous and modified by external ionic composition. *Plant Cell Environ.* 31, 1063-1073.

Hao, L.H., Wang, W.X., Chen, C., Wang, Y.F., Liu, T., Li, X., and Shang, Z.L. (2012). Extracellular ATP promotes stomatal opening of *Arabidopsis thaliana* through heterotrimeric G protein alpha subunit and reactive oxygen species. *Mol. Plant* 5, 852-864.

He, Y., Wu, J., Lv, B., Li, J., Gao, Z., Xu, W., Baluska, F., Shi, W., Shaw, P.C., and Zhang, J. (2015). Involvement of 14-3-3 protein GRF9 in root growth and response under polyethylene glycol-induced water stress. *J. Exp. Bot.* 66, 2271-2281.

Xu, W., Shi, W., Jia, L., Liang, J., and Zhang, J. (2012). TTF6 and TTF7, two different members of tomato 14-3-3 gene family, play distinct roles in plant adaptation to low phosphorus stress. *Plant Cell Environ.* 35, 1393-1406.

REVIEWERS' COMMENTS:

Reviewer #1 (Remarks to the Author):

Review on the revised manuscript by Yang et al entitled 'Calcium-activated 14-3-3 proteins as a molecular switch in salt stress tolerance'

In their revised version of the manuscript the authors have carefully answered the questions and comments raised. In addition, they have included additional experiments as suggested to substantiate their conclusions.

Only minor comments are left:

1. Identification of SOS2-interacting proteins

Q/C: The MS data of other interacting PKS5 proteins should be shown as suppl. information/table (see initial comment).

2. P4/I9: '1b). S_{Ca}BP1, a SOS3-like Calcium Binding Protein, which binds to Ca²⁺ and activates PKS5 to repress PM H⁺ -ATPase activity in yeast.'

Q/C: Should this be a heading? otherwise rephrase.

3. Q/C: Did the densitometric analysis take into account the loading controls (correction factor) or did it just reflect the density of bands in question?

4. Q/C: Suppl. figure 2 – data should be cited in the correct order in text or the sub-figures should be re-arranged.

5. Q/C: Suppl. Fig. 2 – the use of so many different NaCl concentrations should be explained for clarity.

6. Q/C: Non-invasive Micro-test Technology (NMT) measurements – why analysing the root meristematic zone when SOS1 expression is highest in root hair zone?

7. Suppl. Fig. 4 – Salt and Alkali Sensitive Assays

Q/C: 500mM NaHCO₃ was used to adjust pH! Isn't this salt stress, too??

8. Fig. 7f – please use same scaling as in 7a

9. P14/I3: '...the repression of PKS5 activity that reduces SOS2Ser294 phosphorylation...;

Q/C: a possible involvement of phosphatases should be discussed.

Reviewer #2 (Remarks to the Author):

The authors have performed the experiments that I asked. Not all results were easy to fit into the conclusion but it was because of complexity of living systems. I think that no further experiment can be asked in the current project and overall the results contain interesting findings.

minor point: "CCB" (P29,L22) should be "CBB".

Response to Reviewers' Comments

We would like to thank the reviewers again for their helpful evaluation of and comments and suggestions for our manuscript.

All comments have been addressed and additional modifications and explanations have been provided to strengthen our conclusions. These changes have allowed us to substantially improve the clarity of our manuscript for NC readers.

Below, we respond to all Reviewers' comments and questions. The reviewer comments are laid out below in thickened and italicized font and specific concerns have been numbered. Our response is given in normal font.

Response to reviewer 1:

1. Identification of SOS2-interacting proteins

Q/C: The MS data of other interacting PKS5 proteins should be shown as suppl. information/table (see initial comment).

Response:

As suggested, we added the complete mass spectrometry data in Supplementary Table 1.

2. P4/19: '1b). ScaBP1, a SOS3-like Calcium Binding Protein, which binds to Ca²⁺ and activates PKS5 to repress PM H⁺-ATPase activity in yeast.'

Q/C: Should this be a heading? otherwise rephrase.

Response:

As suggested, we corrected this mistake.

3. Q/C: Did the densitometric analysis take into account the loading controls (correction factor) or did it just reflect the density of bands in question?

Response:

Yes, we take into account the loading controls in the densitometric analysis. The value represents the ratio of density of indicated band and the corresponding loading control, for example in Figure 2h, the numbers represent the density of anti-P-SC8 band to anti-C-Myc band.

4. Q/C: Suppl. figure 2 – data should be cited in the correct order in text or the sub-figures should be re-arranged.

Response:

As suggested, we re-arranged the sub-figures in Supplementary Fig. 2.

5. Q/C: Suppl. Fig. 2 – the use of so many different NaCl concentrations should be explained for clarity.

Response:

Expression of the Ser²⁹⁴-to-Asp mutant form (*SOS2*^{T168D/Δ308/S294D}), which enhances the repression of SOS2 kinase activity by 14-3-3s and strongly repressed yeast growth, so we used a lower concentration of NaCl in yeast system to detect the effect of 14-3-3s on the repression of *SOS2*^{T168D/Δ308/S294D} activity (Supplementary Fig. 2g), and the effect of 14-3-3-mediated repression of *SOS2*^{S294A} was relatively weak, so we used a higher concentration of NaCl in yeast system to compare the effect of 14-3-3-mediated repression of *SOS2* and *SOS2*^{S294A} (Supplementary Fig. 2f).

6. Q/C: Non-invasive Micro-test Technology (NMT) measurements – why analysing the root meristematic zone when *SOS1* expression is highest in root hair zone?

Response:

We have measured the Na⁺ flux in root hair zone before, but it was not very stable compared with in root meristematic zone, so we measured the Na⁺ flux in root meristematic zone.

7. Suppl. Fig. 4 – Salt and Alkali Sensitive Assays

Q/C: 500mM NaHCO₃ was used to adjust pH! Isn't this salt stress, too??

Response:

We adjusted the MS medium with 500mM NaHCO₃, the addition of NaHCO₃ was nearly 1.5 mL per liter of MS medium, and the final concentration of Na⁺ did not exceed 1 mM, so it was not salt stress.

8. Fig. 7f – please use same scaling as in 7a

Response:

We adjusted the scaling of Fig. 7f in the revised manuscript.

9. P14/l3: ‘...the repression of *PKS5* activity that reduces *SOS2*Ser294 phosphorylation...;

Q/C: a possible involvement of phosphatases should be discussed.

Response:

Thank you for your suggestion. Previous studies reported that a member of PP2Cs AHG3 interacted with SnRK3.22 (*PKS5*) (Lumba et al., 2014), which may participate in the regulation of *PKS5* kinase activity, and we added it in the revised manuscript.

Lumba, S., Toh, S., Handfield, L.F., Swan, M., Liu, R., Youn, J.Y., Cutler, S.R., Subramaniam, R., Provat, N., Moses, A., et al. (2014). A mesoscale abscisic acid hormone interactome reveals a dynamic signaling landscape in Arabidopsis. *Dev. Cell* 29, 360-372.

Response to reviewer 2:

1. minor point: “CCB” (P29,L22) should be “CBB”.

Response:

We apologized for this mistake and thanks for your correction. We corrected it in the revised manuscript.